# 3D STREETUNVEILER WITH SEMANTIC-AWARE 2DGS - A SIMPLE BASELINE

**Jingwei Xu**[1]**, Yikai Wang**[2*]**, Yiqun Zhao**[3,6]**, Yanwei Fu**[5†]**, Shenghua Gao**[3,4‡]
[1] ShanghaiTech University  [2] Nanyang Technological University  [3] The University of Hong Kong
[4] HKU Shanghai Intelligent Computing Research Center  [5] Fudan University  [6] Transcengram
xujw2023@shanghaitech.edu.cn   yikai.wang@ntu.edu.sg
yiqun.zhao@connect.hku.hk   yanweifu@fudan.edu.cn   gaosh@hku.hk

## ABSTRACT

Unveiling an empty street from crowded observations captured by in-car cameras is crucial for autonomous driving. However, removing all temporarily static objects, such as stopped vehicles and standing pedestrians, presents a significant challenge. Unlike object-centric 3D inpainting, which relies on thorough observation in a small scene, street scene cases involve long trajectories that differ from previous 3D inpainting tasks. The camera-centric moving environment of captured videos further complicates the task due to the limited degree and time duration of object observation. To address these obstacles, we introduce StreetUnveiler to reconstruct an empty street. StreetUnveiler learns a 3D representation of the empty street from crowded observations. Our representation is based on the hard-label semantic 2D Gaussian Splatting (2DGS) for its scalability and ability to identify Gaussians to be removed. We inpaint rendered image after removing unwanted Gaussians to provide pseudo-labels and subsequently re-optimize the 2DGS. Given its temporal continuous movement, we divide the empty street scene into observed, partial-observed, and unobserved regions, which we propose to locate through a rendered alpha map. This decomposition helps us to minimize the regions that need to be inpainted. To enhance the temporal consistency of the inpainting, we introduce a novel time-reversal framework to inpaint frames in reverse order and use later frames as references for earlier frames to fully utilize the long-trajectory observations. Our experiments conducted on the street scene dataset successfully reconstructed a 3D representation of the empty street. The mesh representation of the empty street can be extracted for further applications.

## 1 INTRODUCTION

Accurate 3D reconstruction of an empty street scene from an in-car camera video is crucial for autonomous driving. It provides reliable digital environments that simulate real-world street scenarios. Although this is an important task, it is seldomly studied in previous works because of its challenging nature in the following aspects: (1) Lack of ground truth data for pre-training inpainting models specialized for street scenes; (2) The camera-centric moving captures objects from limited angles and for brief periods; (3) The long trajectory of in-car camera videos leads to objects appearing and disappearing at different time points, complicating object removal.

But there still exists a blessing we can take from the long trajectory moving-forward nature. As the car moves forward, objects that disappear from the later frame will only be visible in previous video frames. This gives a hint about maintaining the temporal consistency of the same regions.

To address the challenge of reconstructing an empty street, we introduce **StreetUnveiler**, a reconstruction method targeting unveiling the empty representation of long-trajectory street scenes.

---

[*]Yikai Wang was affiliated with Fudan University during the development of this paper.

[†]Prof. Yanwei Fu is also with Fudan ISTBI–ZJNU Algorithm Centre for Brain-inspired Intelligence, Zhejiang Normal University, Jinhua, China.

[‡]Corresponding author

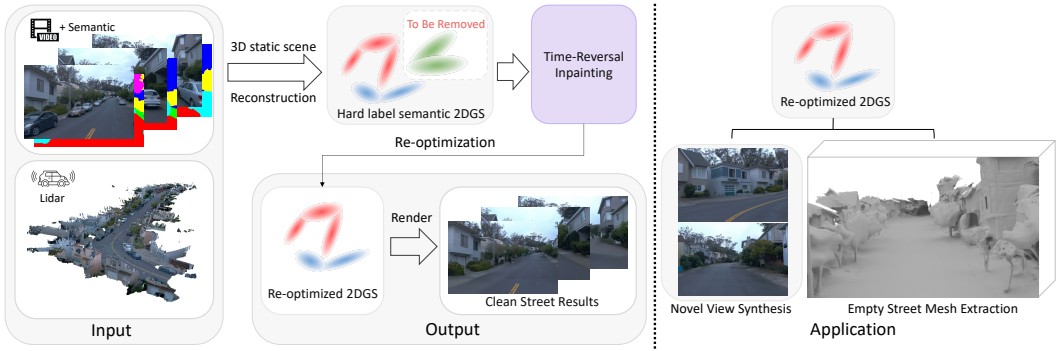

Figure 1: We achieve accurate empty street reconstruction from in-car camera videos. With the aid of the proposed hard-label semantic 2D Gaussian Splatting and time-reversal inpainting framework, we remove the unwanted objects with satisfactory appearance and geometry of occluded regions.

StreetUnveiler involves several key steps. First, it reconstructs the observed 3D representation and identifies unobserved regions occluded by objects. Then, it uses a time-reversal inpainting framework to consistently inpaint these unobserved regions as pseudo labels. Finally, it re-optimizes the 3D representation based on these pseudo labels. The overall pipeline is illustrated in Fig. 1.

StreetUnveiler first reconstructs the original parked-up street with Gaussian Splatting (GS) due to its scalability and editability. However, as is illustrated in Fig. 2, inpainting with the naïve object mask (orange mask) often results in blurring and loss of details in large inpainted regions, which is a common issue in the previous works Mirzaei et al. (2023); Weder et al. (2023); Wang et al. (2023a); Weber et al. (2024); Liu et al. (2024). Generating masks for completely unobservable regions (blue mask) that are invisible from any viewpoint remains a challenge. Recent work Liu et al. (2024) requires user-provided masks, which is impractical for long trajectories. Moreover, the messy appearance of these regions after removing the Gaussians makes it difficult to use methods like SAM Kirillov et al. (2023). To address the difficulty of finding an ideal inpainting mask, we propose to generate the mask through the rendered alpha map and reconstruct the scene using a hard-label semantic 2DGS Huang et al. (2024a) instead of 3DGS Kerbl et al. (2023) . 2DGS has a high opacity value for Gaussians, resulting in low alpha values in completely unobservable regions. A semantic distortion loss and a shrinking loss are employed to further reduce the rendered alpha values of the completely unobservable regions. This approach automatically generates masks for unobservable regions without user input, leading to better inpainting results.

Furthermore, we propose a time-reversal inpainting framework to enhance the temporal consistency of inpainting results in completely unobservable regions. By inpainting the video frames in reverse order, we use the later frame as a reference to inpaint the earlier frame. When the video is played in reverse, the object in the later frame will transition only from near to far in the camera view as the camera moves away from the object in a reversed time-space. This method uses a high-to-low-resolution guiding approach instead of filling an area larger than the reference region, as in the low-to-high-resolution approach, which results in more consistent inpainting. Finally, the inpainted pixels are used as pseudo labels to guide the re-optimization of 2DGS. This enables our method to learn a scalable 2DGS model that represents an empty street while preserving the appearance integrity of regions visible in other views.

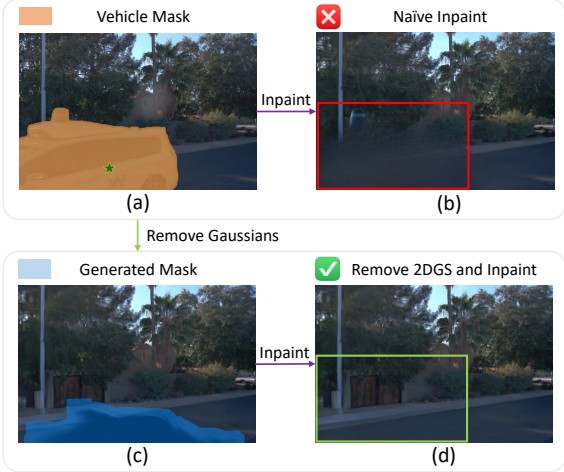

Figure 2: (a) The mask of the whole unwanted object; (b) Inpainting with (a) mask; (c) Generate the inpainting mask through a rendered alpha map. The pixel with a low alpha value is selected as an inpainted pixel; (d) Inpainting with the generated (c) mask.

Our contribution can be summarized as follows:

- We propose representing the street as hard-label semantic 2DGS, optimizing the 3D scene with semantic guidance for scalable representation and improved instance decoupling.
- We use a rendered alpha map to locate completely unobservable regions and apply a semantic distortion loss and a shrinking loss to create a reasonable inpainting mask for these regions.
- We introduce a novel time-reversal inpainting framework for long-trajectory scenes, enhancing the temporal consistency of inpainting results for re-optimization. Experiments show that our method can reconstruct an empty street from in-car camera video containing obstructive elements.

## 2 RELATED WORK

**Neural scene representation and reconstruction.** The use of neural radiance fields (NeRF)Mildenhall et al. (2020) to represent 3D scenes inspired a lot of follow-up work based on the original approach. Some worksMüller et al. (2022); Chen et al. (2022); Sun et al. (2022); Sara Fridovich-Keil and Alex Yu et al. (2022); Yu et al. (2022); Xiao et al. (2024) explore explicit representations such as low-rank matrices, hash grids, or voxel grids to increase the model capacity of original MLPs. Some works explored multiple separate MLPs Reiser et al. (2021); Kundu et al. (2022); Fu et al. (2022) to represent instances and backgrounds separately. However, these scale-up strategies are complicated to implement at the scale of street scenes. Existing works Xu et al. (2023); Tancik et al. (2022); Turki et al. (2022); Lu et al. (2023); Rematas et al. (2022); Yang et al. (2023b); Wang et al. (2023c); Turki et al. (2023); Meuleman et al. (2023); Wang et al. (2023b); Guo et al. (2023); Zhang et al. (2023b); Siddiqui et al. (2023) explored mesh-based, primitive-based, or grid-based representations for large-scale street scenes. However, both grid-based representation Guo et al. (2023) and mesh-based representation Wang et al. (2023c) may be constrained by their limited topology, making it hard to decouple the scene into separate instances. Recent advances in point-based rendering techniques Kerbl et al. (2023); Lassner & Zollhofer (2021); Xu et al. (2022); Huang et al. (2024a) can achieve both high-quality and fast rendering speed. The point-based nature of Gaussian Splatting enables scalability for street scenes. While recent works Chen et al. (2023b); Yan et al. (2024); Lin et al. (2024b); Ren et al. (2024); Cheng et al. (2024) have explored the reconstruction of large-scale scenes using Gaussian Splatting, our work focuses on the unveiling stage of a street scene, which is more important for autonomous driving and more challenging.

**3D scene manipulation and inpainting.** Early works Wang et al. (2021); Yuan et al. (2020); Philip & Drettakis (2018); Thonat et al. (2016); Anguelov et al. (2010); Liu et al. (2018); Yu et al. (2018; 2019); Yi et al. (2020); Zhao et al. (2021); Mirzaei et al. (2024) explored street scene editing by leveraging single-view or multi-view image inpainting networks. With the rapid development of Neural Scene Representation, editing a 3D scene has been explored by lots of works Chong Bao and Bangbang Yang et al. (2022); Zhao et al. (2024b); Yang et al. (2023a); Yuan et al. (2022); Bao et al. (2023); Kobayashi et al. (2022); Kerr et al. (2023); Peng et al. (2023); Zhao et al. (2024a). Edit-NeRF Liu et al. (2021) pioneered shape and color editing of neural fields using latent codes. Subsequent works Bao et al. (2023); Kobayashi et al. (2022); Kerr et al. (2023); Peng et al. (2023) utilized CLIP models to provide editing guidance from text prompts or reference images. Recent works Weder et al. (2023); Zhang et al. (2022); Mirzaei et al. (2023); Xiang et al. (2023); Chen et al. (2023a); Fang et al. (2023); Ye et al. (2023); Weber et al. (2024); Wang et al. (2024c); Lin et al. (2024a); Mirzaei et al. (2024); Prabhu et al. (2023) also explored 2D stylization and inpainting techniques, utilizing pretrained Diffusion Priors Rombach et al. (2022) for editing 3D scenes. Specifically, Chen et al. (2023a); Fang et al. (2023); Ye et al. (2023); Wang et al. (2024c) investigated these approaches in collaboration with Gaussian Splatting. Unlike them, our work focuses on street scene object removal and empty street reconstruction, which is more challenging.

**Image and video inpainting**. Image inpainting Bertalmio et al. (2000) aims to fulfill the missing region within an image. Standard approaches included GAN-based methods Pathak et al. (2016); Zhao et al. (2020), attention-based methods Yu et al. (2018); Liu et al. (2019), transformer-based methods Wan et al. (2021); Liu et al. (2022), and more recently, diffusion-based methods Rombach et al. (2022); Wang et al. (2024a). Control-Net Zhang et al. (2023a) enabled generating images with additional conditions on the frozen diffusion models. Recently, LeftRefill Cao et al. (2024) learned to guide the frozen diffusion inpainting models with extra conditions of the reference image, enabling

multi-view inpainting on the frozen diffusion model. However, these image inpainting methods mainly focused on the static scenario. Video inpainting considers the temporal consistent inpainting in the continuous image sequence, utilizing approaches like 3D CNN Wang et al. (2019); Hu et al. (2020), temporal shifting Zou et al. (2021), flow guidance Kim et al. (2019); Xu et al. (2019); Li et al. (2022), temporal attentions Ren et al. (2022), to name a few. However, these video inpainting methods hardly considered the long trajectory movement of cameras. In contrast, in our paper, we focus on the inpainting of large-scale street scenes. Furthermore, the 2DGS representation used in our paper enables the free-view rendering of the inpainted video.

## 3 PROBLEM FORMULATION

Given in-car camera videos and the Lidar data of a parked-up street, our goal is to remove all temporarily static objects in the street, like stopping vehicles and standing pedestrians, and finally reconstruct an empty street. This task, named as **Street Unveiling**, is to reconstruct scenes devoid of these static obstacles, providing an empty representation of the street environment. Such representations are mainly represented by 3D models for free-view rendering. This task holds significant implications for autonomous driving systems, urban planning, and scene understanding applications.

Street Unveiling shares some similarities with related tasks but cannot be addressed using existing approaches. (1) 3D reconstruction primarily involves modeling a primary image or scene with an object-centric camera. In contrast, Street Unveiling focuses on the background, aiming to remove foreground objects to reveal an empty street. The absence of ground truth further differentiates it from standard 3D reconstruction tasks. (2) Video inpainting typically deals with videos captured by fixed or minimally moving cameras, featuring one or a few central objects. Conversely, Street Unveiling involves long camera trajectories without central objects. These distinctions require different capabilities and novel methods to address the unique challenges of Street Unveiling.

## 4 SEMANTIC STREET RECONSTRUCTION

We opt for 2D Gaussian Splatting Huang et al. (2024a) (2DGS) as our scene representation for its rendering speed and editability. We first introduce the 2DGS in Sec. 4.1. Subsequently, we elaborate our algorithm tailored for street unveiling using 2DGS in Sec. 4.2 and Sec. 4.3.

### 4.1 PRELIMINARY: 2D GAUSSIAN SPLATTING

Our reconstruction stage builds upon the state-of-the-art point-based renderer with the splendid geometry performance, 2DGS Huang et al. (2024a). 2DGS is defined by several key components: the central point $\mathbf{p}_k$, two principal tangential vectors $\mathbf{t}_u$ and $\mathbf{t}_v$ that determine its orientation, and a scaling vector $\mathbf{S} = (s_u, s_v)$ controlling the variances of the 2D Gaussian distribution.

2D Gaussian Splatting represents the scene's geometry as a set of 2D Gaussians. A 2D Gaussian is defined in a local tangent plane in world space, parameterized as follows:

$$P(u, v) = \mathbf{p}_k + s_u \mathbf{t}_u u + s_v \mathbf{t}_v v. \tag{1}$$

For the point $\mathbf{u} = (u, v)$ in $uv$ space, its 2D Gaussian value can then be evaluated using the standard Gaussian function:

$$\mathcal{G}(\mathbf{u}) = \exp\left(-\frac{u^2 + v^2}{2}\right). \tag{2}$$

The center $\mathbf{p}_k$, scaling $(s_u, s_v)$, and the rotation $(\mathbf{t}_u, \mathbf{t}_v)$ are learnable parameters. Each 2D Gaussian primitive has opacity $\alpha$ and view-dependent appearance $\mathbf{c}$ with spherical harmonics. For volume rendering, Gaussians are sorted according to their depth value and composed into an image with front-to-back alpha blending:

$$\mathbf{c}(\mathbf{x}) = \sum_{i=1} \mathbf{c}_i \alpha_i \mathcal{G}_i(\mathbf{u}(\mathbf{x})) \prod_{j=1}^{i-1} (1 - \alpha_j \mathcal{G}_j(\mathbf{u}(\mathbf{x}))). \tag{3}$$

where $\mathbf{x}$ represents a homogeneous ray emitted from the camera and passing through $uv$ space.

## 4.2 2DGS FOR STREET SCENE RECONSTRUCTION

2DGS Huang et al. (2024a) features for its accurate geometry reconstruction of the object surface. However, the application of 2DGS to reconstruct objects devoid of surfaces, such as the sky in an open-air street scene, remains unexplored. We aim to reconstruct the street scene as a radiance field and semantic field using 2DGS. More details about radiance field reconstruction are included in the supplementary.

**Learning 2D Gaussians with semantic guidance.** We aim to augment the radiance field of street scenes with editability. Inspired from Guo et al. (2022); Yan et al. (2024); Chen et al. (2023b); Zhou et al. (2024), we harness the power of 2D semantic segmentation and distill such knowledge back to 2D Gaussians. To do so, we inject each 2D Gaussian with a 'hard' semantic label. The 'hard' means that the semantic label is non-trainable, which differs from the learnable 'soft' label used in recent works Zhou et al. (2024); Yan et al. (2024); Zhou et al. (2023b). Note that although our 'hard' semantic label is not trainable, it allows for rendering correct 2D semantic maps by altering its opacity, rotation, scaling, and position. This encourages the points with the same semantic labels to gather closer, facilitating accurate object removal in 3D space. Assume that each 2D Gaussian associated with a one-hot encoded semantic label s, we render the 2D semantic map as:

$$\hat{S}(\mathbf{x}) = \sum_{i=1} \mathrm{s}_i \alpha_i \mathcal{G}_i(\mathbf{u}(\mathbf{x})) \prod_{j=1}^{i-1} (1 - \alpha_j \mathcal{G}_j(\mathbf{u}(\mathbf{x}))). \tag{4}$$

During our densification, the newly generated splats will inherit the original hard semantic labels.

## 4.3 OPTIMIZATION OF 2DGS FOR STREET UNVEILING

In this part, we first introduce standard objectives used by previous approaches to optimize 2DGS Huang et al. (2024a). Then we discuss the inferiority of these objectives in the street scene and propose the newly introduced objectives tailored for Street Unveiling. In summary, our objectives consist of photo-metric loss, semantic loss, normal consistency loss, two different depth distortion losses, and shrinking loss.

**Standard approach**: As in 3DGS Kerbl et al. (2023), we use $\mathcal{L}_1$ loss and D-SSIM loss for supervising RGB color, with $\lambda = 0.2$:

$$\mathcal{L}_{\mathrm{rgb}} = (1 - \lambda)\mathcal{L}_1 + \lambda \mathcal{L}_{\text{D-SSIM}}. \tag{5}$$

Following 2DGS Huang et al. (2024a), depth distortion loss and normal consistency loss are adopted to refine the geometry property of the 2DGS representation of the street scene.

$$\mathcal{L}_{\mathrm{d}} = \sum_{i,j} \omega_i \omega_j |z_i - z_j| \qquad \mathcal{L}_n = \sum_i \omega_i (1 - \mathbf{n}_i^\top \mathbf{N}) \tag{6}$$

Here, $\omega_i$ represents the blending weight of the $i-$th intersection. $z_i$ denotes the depth of the intersection points. $\mathbf{n}_i$ is the normal of the splat facing the camera. $\mathbf{N}$ is the estimated normal at nearby depth point $\mathbf{p}$.

We employ Cross-Entropy (CE) loss to supervise semantic labels:

$$\mathcal{L}_s(\mathbf{x}) = \mathrm{CE}(\hat{S}(\mathbf{x}), S(\mathbf{x})) \tag{7}$$

where $S$ is a pseudo semantic map extracted from a pre-trained segmentation model Xie et al. (2021).

**Inferiority of standard objectives**. In Street Unveiling, the scene semantics are expected to be maintained in a less messy and more consistent way to better recognize the Gaussians of objects to remove. However, solely naïve depth distortion won't hinder the merging of the 2DGS Huang et al. (2024a) with different semantic labels, leading to noisy semantic information about the 3D world. Meanwhile, the noisy Gaussians in the unseen region will still exist if we don't find a way to eliminate them. These problems will further harm the generation of an ideal inpainting mask.

**Clean up objectives**. To reduce the noise in the semantic fields, we propose a semantic depth distortion loss $\mathcal{L}_{\mathrm{ds}}$ and a shrinking loss $\mathcal{L}_\alpha$ on opacity $\alpha$:

$$\mathcal{L}_{\mathrm{ds}} = \sum_k \mathcal{L}_{\mathrm{d}}^k \qquad \mathcal{L}_\alpha = \frac{1}{N} \sum_p \alpha_p \tag{8}$$

where $k$ iterates over each semantic label and $\mathcal{L}_{\mathrm{d}}^k$ denotes the distortion loss of 2DGS Huang et al. (2024a) with same semantic labels. This semantic depth distortion loss is exerted on the rendered result of the Gaussians with the same semantic label. Intuitively, it will encourage the 2DGS with the same label to have a more consistent depth at the pixel level. Shrinking loss will further eliminate the Gaussians that are actually unseen by any viewpoint. $\alpha_p$ represents the opacity value $\alpha$ of each Gaussian. $N$ is the total number of Gaussians.

The total loss is given as

$$\mathcal{L} = \mathcal{L}_{\mathrm{rgb}} + \lambda_d \mathcal{L}_{\mathrm{d}} + \lambda_n \mathcal{L}_{\mathrm{n}} + \lambda_{ds} \mathcal{L}_{\mathrm{ds}} + \lambda_s \mathcal{L}_{\mathrm{s}} + \lambda_\alpha \mathcal{L}_\alpha \tag{9}$$

We empirically set $\lambda_d = 100$, $\lambda_n = 0.05$, $\lambda_{ds} = 100$, $\lambda_s = 0.1$, and $\lambda_\alpha = 0.001$.

## 5 EMPTY STREET RECONSTRUCTION

A common strategy Mirzaei et al. (2023); Weder et al. (2023) to inpaint within small scenes is utilizing 2D inpainting methods to inpaint removed objects in the image space for re-optimization. However, lots of problems arise when it comes to the street scene. (1) Some views result in over-blurry inpainting results due to the huge size of the inpainting mask, as is illustrated in Fig. 2(b); (2) Some occluded regions of the street struggle to maintain consistency because they are exposed to a large number of views over the long trajectory. These challenges will make it more vulnerable to inconsistent inpainting.

In the context of point-based scene representation, eliminating the object involves deleting Gaussians. However, a naïve removal often yields unsatisfactory results, particularly in the completely unobservable regions beneath the object. In this section, we first propose how to generate the ideal mask for inpainting as in Fig. 2(c). Then, we propose our time-reversal inpainting framework and how to use the inpainting results to re-optimize the 2DGS.

### 5.1 GENERATION OF IDEAL INPAINTING MASK

In the street video captured by a moving car, we can divide the pixel space into three categories: (1) The observable regions, where the regions are not occluded by any objects; (2) The partially observable regions, where the regions are occluded in some views but are observable in other views; (3) The completely unobservable regions, where the regions are unobservable in all recording views. For regions in the second case, we can utilize information from other views to preserve more information about the appearance of the street scene. As illustrated in Fig. 2, naïvely inpainting with the object mask will cause the unexpected blurry inpainting result at the partially observable region, which can be viewed from other viewpoints but is occluded from the current viewpoint.

To distinguish partially observable regions from completely unobservable regions and improve the inpainting quality, we propose using the rendered alpha map to generate the mask for completely unobservable regions. For a given viewpoint, we first remove the Gaussians of unwanted objects. For robustness, we remove other Gaussians that are "too close" to the previously removed Gaussians. Then we render the alpha map of the remaining scene. We identify the completely unobservable region via pixels with low alpha values. The pixels with alpha values lower than a threshold are selected as inpainting masks. The threshold is set as 0.99 in our implementation.

### 5.2 TIME-REVERSAL INPAINTING

The core challenge in reconstructing the empty street scene is ensuring consistency between different views over the long trajectory. However, current video inpainting methods cannot generalize to our long trajectory and complex scenarios, which can be validated from Tab. 1, Fig. 5, and supplementary video comparison. This usually lags behind the scale-up speed of image inpainting models. To this end, we propose using a reference-based image inpainting method that is trained to ensure consistency between the inpainted region and the reference-based image. Particularly, we adopt the LeftRefill Cao et al. (2024) for its stable diffusion-based backbone and the matching-based training strategy. The stable diffusion backbone leads to a more powerful inpainting model with a strong generation capacity in open-world scenarios, which fits the requirement of Street Unveiling. Furthermore, the matching-based training strategy ensures that the inpainting model correctly fulfills

the masked region based on the observation in the reference image, which encourages consistency between different views.

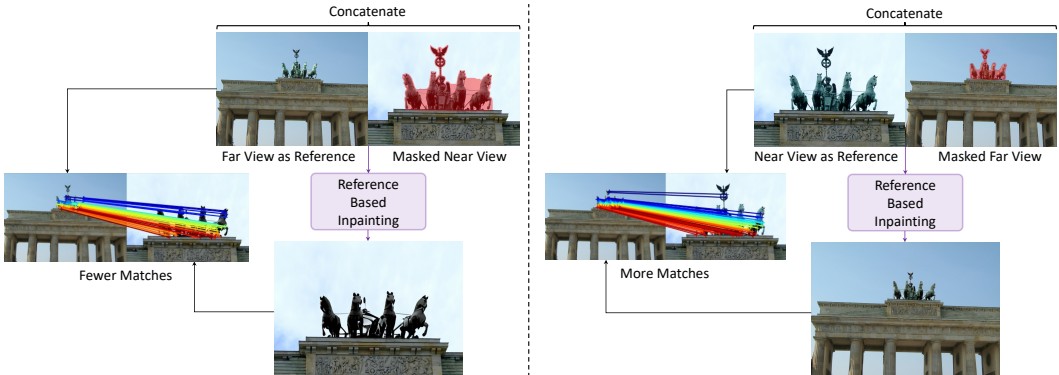

Figure 3: Illustration of reference-based inpainting of two views. **Left:** When we inpaint the near view with the far view as a reference, the consistency of the inpainting result degenerates. There are fewer matching pixels between the reference far-view image and near-view inpainting result; **Right:** Inpainting the far view using the near view as a reference results in better quality and more accurate pixel matching. It's easier to generate the low-resolution content with the high-resolution image as a reference.

However, time-forward inpainting sequences usually lead to the failure of consistent inpainting. Given the moving-forward nature of data-collecting vehicles, objects to be removed transit from far to near in the camera view. **(1)** As is illustrated in Fig. 3, when we use the far-view image as a reference to inpaint the same region in the near-view image, the models may not correctly capture the matching relationships and thus causing inconsistent inpainting. Conversely, setting the near-view image as the reference image leads to a more precise matching result and naturally better inpainting results. **(2)** The near-view image can capture more fine-grained information and a larger receptive field, thus making the inpainting easier to inpaint in a high-to-low resolution instead of low-to-high which requires extra super-resolution capacity for the inpainting model. Besides, the objects removed in the final frame are consistently observed in the earlier frames.

Based on the above analysis, we propose the time-reversal inpainting framework. If we reverse the time, we can turn the moving-forward nature into a moving-backward nature. When the time is reversed, objects to be removed will instead transition from near to far in the camera view because the camera will be away from the removed object in reversed time-space.

We target to unconditionally inpaint a 3D region only once and then transmit the inpainted region's pixels to other views with reference-based inpainting. As is illustrated in Fig. 4, we first unconditionally inpaint both frame $T_n$ and $T_{n+1}$ with Cao et al. (2023). However, for frame $T_n$, there are some regions that can be seen in $T_{n+1}$. We expect they would share more matching pixels by utilizing the implicit pixel-matching ability of reference-based inpainting model Cao et al. (2024). Then we use frame $T_{n+1}$ as a reference to inpaint frame $T_n$, masking only the regions visible in $T_{n+1}$. More implementation details are elaborated in Sec. A.2 of the supplementary.

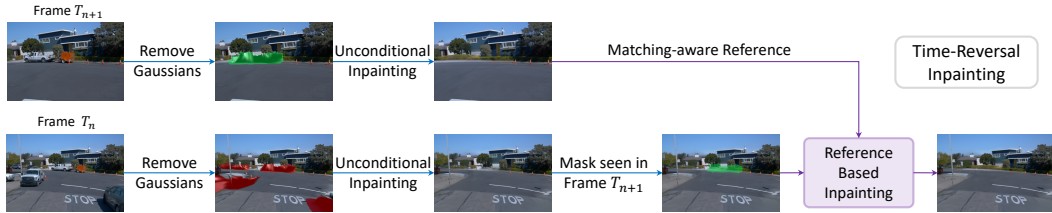

Figure 4: Illustration of time-reversal inpainting. After we remove the Gaussians of the objects, we first unconditionally inpaint both frame $T_n$ and $T_{n+1}$ with Cao et al. (2023). Then we transmit the pixels from frame $T_{n+1}$ to frame $T_n$ in the form of reference-based inpainting Cao et al. (2024). From a high-level understanding, we inpaint the earlier frame $T_n$ with the later frame $T_{n+1}$ as a reference condition.

## 5.3 RE-OPTIMIZATION OF THE 2D GAUSSIANS

Once we finish time reversal inpainting, we use our inpainting results as pseudo labels to guide the re-optimization of 2DGS Huang et al. (2024a). We use the following loss for re-optimization:

$$\mathcal{L}_{\text{retrain}} = \mathcal{L}_1 + \lambda_d \mathcal{L}_{\text{d}} + \lambda_n \mathcal{L}_{\text{n}}. \tag{10}$$

## 6 EXPERIMENTS

Our experiments were conducted on a single NVIDIA A40 GPU with peak memory usage of 16GB.

**Dataset.** For the evaluation of our approach from the reconstruction aspect and the object removal aspect, we adopt real-world street scenes from Waymo Open Perception Dataset Sun et al. (2020) and Pandaset Xiao et al. (2021). The Waymo dataset collects data from 5 camera perspectives, encompassing roughly 230 degrees in field of view (FOV). We downscale the resolution to $484 \times 320$. The Pandaset collects data from 6 camera perspectives, encompassing 360 degrees in FOV. We downscale the resolution to $480 \times 270$. We select front-view video sequences as the same experimental setup in Yan et al. (2024); Chen et al. (2023b); Zhou et al. (2024), using 24 scenes from Waymo and 9 scenes from Pandaset for our experiments.

**Metrics.** To evaluate the effectiveness of object removal, we approach it from a multi-view inpainting perspective. We follow the well-established previous works Mirzaei et al. (2023); Weder et al. (2023); Liu et al. (2024); Lin et al. (2024a), we calculate the LPIPS Zhang et al. (2018) and Fréchet Inception Distance (FID) Heusel et al. (2017) scores to quantify the discrepancies between the ground-truth views and removed results. Each output video frame is paired with the corresponding frame from the original training video to compute the LPIPS. We use the image collections of the output video and original training video to compute FID.

**Baselines.** We compare our approach to 3D inpainting method SPIn-NeRF Mirzaei et al. (2023) and a recent Gaussian Splatting based inpainting method Infusion Liu et al. (2024). As the original MLP implementation of SPIn-NeRF Mirzaei et al. (2023) works poorly in the large-scale street scene, we re-implement SPIn-NeRF Mirzaei et al. (2023) based on 2DGS Huang et al. (2024a), clarifying that our superiority not only from 2DGS but also the proposed time reversal inpainting. Infusion Liu et al. (2024) is evaluated with the official implementation. Since Infusion Liu et al. (2024) is designed for small scenes, it only conducts GS removal and projection once for the whole scene. Its original setting doesn't match our long-trajectory tasks. Instead, we conduct every 10 frames to fit our setting.

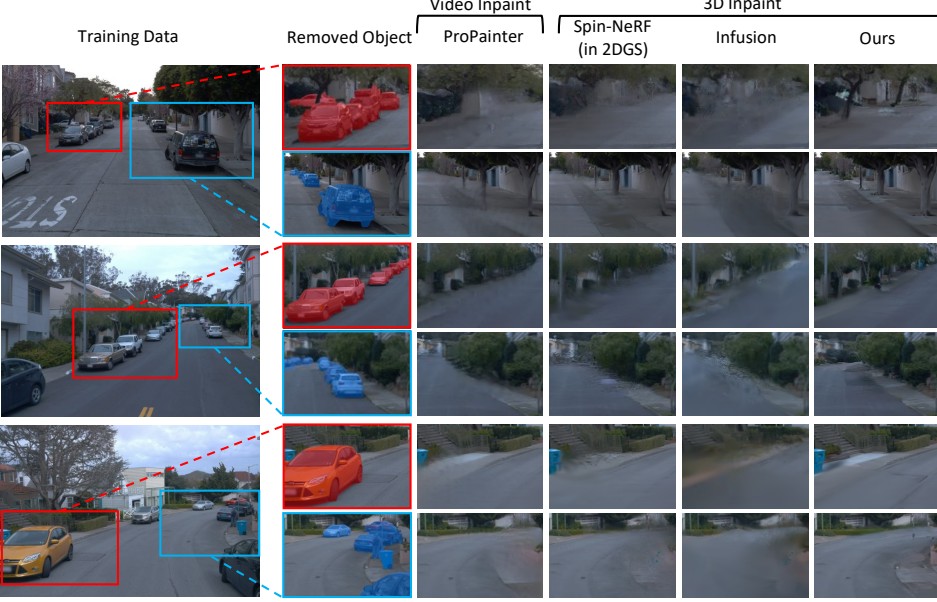

Figure 5: **Qualitative comparison results of our methods.** Our methods achieve clearer results than temporally consistent inpainting baselines. Video comparisons will be placed in the supplementary.

| | Waymo | | Pandaset | |
|---|---|---|---|---|
| | LPIPS↓ | FID ↓ | LPIPS↓ | FID ↓ |
| **Single Image Inpainting** | | | | |
| LaMa(2D) Suvorov et al. (2021) | 0.228 | 138.089 | 0.276 | 160.895 |
| SDXL Podell et al. (2023) | 0.231 | 116.634 | 0.276 | 133.042 |
| **Video Inpainting** | | | | |
| ProPainter Zhou et al. (2023a) | 0.233 | 141.906 | 0.286 | 178.135 |
| **3D Inpainting** | | | | |
| SPIn-NeRF Mirzaei et al. (2023) (in 2DGS) | 0.221 | 140.831 | 0.266 | 174.223 |
| Infusion Liu et al. (2024) | 0.307 | 176.882 | 0.325 | 176.882 |
| Ours | 0.216 | 127.581 | 0.261 | 155.527 |

Table 1: Comparison with state-of-the-art 2D/3D inpainting methods on both two datasets. Our FID is only lower than SDXL, yet SDXL doesn't maintain consistency between different video frames.

| | Waymo | | Pandaset | |
|---|---|---|---|---|
| | LPIPS↓ | FID ↓ | LPIPS↓ | FID ↓ |
| **Ablation of different pseudo labels** | | | | |
| w/LaMa Suvorov et al. (2021) | 0.226 | 137.753 | 0.281 | 169.836 |
| w/SDXL Podell et al. (2023) | 0.222 | 139.716 | 0.279 | 169.805 |
| w/ProPainter Zhou et al. (2023a) | 0.224 | 138.944 | 0.281 | 169.848 |
| **Ablation of 3D representation** | | | | |
| w/3DGS Kerbl et al. (2023) | 0.219 | 140.749 | 0.280 | 161.203 |
| Time-Forward Inpainting | 0.220 | 136.858 | 0.270 | 158.166 |
| Ours | **0.216** | **127.581** | **0.261** | **155.527** |

Table 2: Quantitative ablation study on both two datasets over different 2D inpainting methods for 3D inpainting. And the ablation study over different 3D representations. The comparison verifies the effectiveness of the time-reversal inpainting pipeline and the necessity of the 2DGS representation.

## 6.1 COMPARISON

The quantitative comparison results are shown in Tab. 1, and the qualitative comparison of 3D inpainting methods are shown in Fig. 5. Noticed that SPIn-NeRF Mirzaei et al. (2023) utilizes LaMa Suvorov et al. (2021) and Infusion Liu et al. (2024) utilizes SDXL Podell et al. (2023) for inpainting. We can observe that 3D inpainting baseline methods lead to worse results, especially when the case is challenging. The results demonstrate that our proposed method achieves better 3D inpainting results from the appearance aspect. The geometry property of the removed region will be discussed in the supplementary. Video comparisons will also be included in the supplementary. In Tab. 1, our proposed method outperforms all the baselines in LPIPS. It only achieves a lower FID compared to SDXL, yet SDXL doesn't maintain consistency between different video frames. This can be easily observed from supplementary videos.

## 6.2 FURTHER ANALYSIS

**Ablation of different inpainting methods as pseudo labels.** We compare the reconstruction results with pseudo labels from different inpainting methods. From Fig. 6, we can observe that time reversal will maintain the consistency between View 1 and View 2. Current single image inpainting models, like LaMa Suvorov et al. (2021) and SDXL Podell et al. (2023), fail to maintain the consistency over the video frames. Although the video inpainting models Zhou et al. (2023a) can be temporarily consistent at near frames, the whole inpainting region will be blurred since it can not guarantee 3D consistency. The quantitative results in Tab. 2 verify the effectiveness of our time-reversal pipeline.

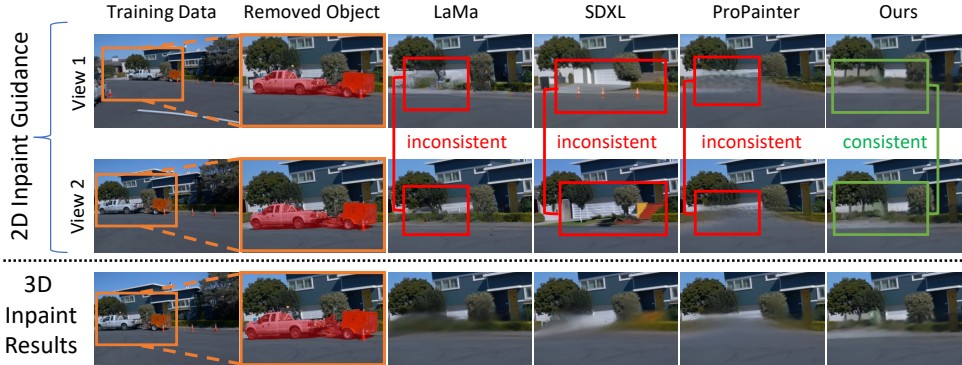

Figure 6: **Ablation for different inpainting methods as pseudo labels.** From the top two rows, we can observe that time-reversal inpainting is able to achieve more consistent inpainting results than other methods. The bottom row shows that our method can achieve better 3D inpainting results.

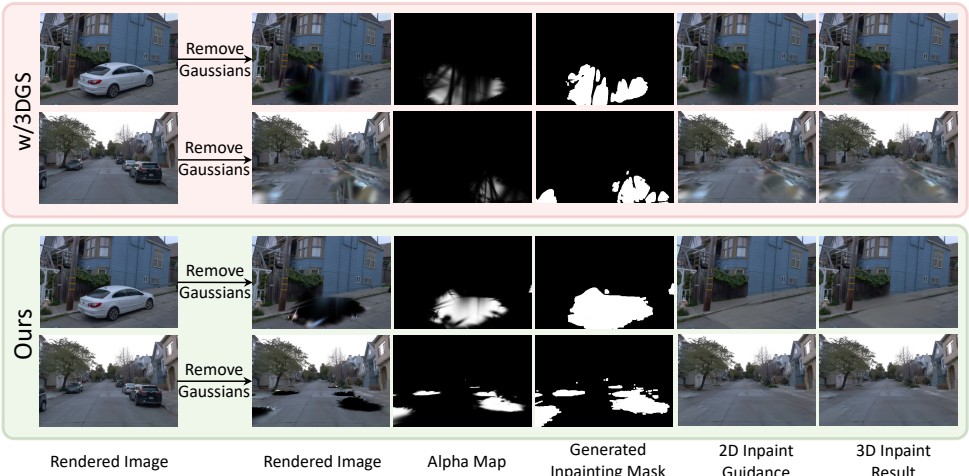

Figure 7: **Ablation for 3D representation.** When we use 3DGS, generating an ideal inpainting mask with a rendered alpha map is hard. Good inpainting results are hard to achieve.

**Ablation of 3D representation.** We ablate through the 3D representation by comparing the results obtained with 3DGS Kerbl et al. (2023) and 2DGS Huang et al. (2024a). From Fig. 7, we can observe that after we remove the Gaussians, the rendered alpha map with 3DGS fails to generate an ideal inpainting mask. The quantitative results given in Tab. 2 verify the necessity of 2DGS representation.

**Ablation of time-reversal inpainting.** For our time-reversal inpainting, we conduct inpainting on frame $T_n$ with $T_{n+1}$ as reference. We additionally ablate the time order in our inpainting progress. For time-forward inpainting, frame $T_{n+1}$ is inpainted with frame $T_n$ as reference. Tab. 2 quantitatively demonstrates the necessity of the time-reversal order. We provide a more comprehensive discussion and qualitative illustration in Sec. B.1 of the supplementary.

## 7 CONCLUSION

We propose StreetUnveiler, a pipeline for reconstructing empty streets from in-car camera videos. Our method represents the street scene using a hard-label semantic-aware 2D Gaussian Splatting Huang et al. (2024a), allowing us to remove each instance from the scene seamlessly. To create an ideal inpainting mask, we utilize the rendered alpha map after removing unwanted 2DGS. Additionally, we introduce a novel time-reversal inpainting framework that enhances consistency across different viewpoints, facilitating the reconstruction of empty streets. Extensive experiments demonstrate that our method effectively reconstructs empty street scenes and supports free-viewpoint rendering.

## 8 ACKNOWLEDGEMENT

The work was supported by NSFC #62172279, #61932020, Program of Shanghai Academic Research Leader. Shuo Wang, Binbin Huang, Zibo Zhao are greatly appreciated for valuable proofreading.

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

# A    IMPLEMENTATION DETAILS

## A.1    DETAILS OF HARD-LABEL SEMANTIC 2DGS RECONSTRUCTION

**Initialization with Lidar points.** High-quality appearance and semantic reconstruction of the whole street scene are hard to reach, with barely SFM points Schönberger & Frahm (2016); Schönberger et al. (2016) as initialization for street scenes. Lidar points are leveraged to better reconstruct the street scene like in Yan et al. (2024); Chen et al. (2023b); Zhou et al. (2024). We use an off-the-self 2D semantic segmenter Xie et al. (2021) to process the 2D images and back-project the hard semantic labels to 2D Gaussians Huang et al. (2024a).

**Environment map for street reconstruction.** We empirically find that most 2D Gaussians' opacity will be larger than $0.9$ or lower than $0.1$, leading to the imperfect reconstruction quality of the background environment, *i.e.*, sky. To better model the environment in the street scene, we employ a tiny MLP $f$ to query the color of the environment map, which is similar to Guo et al. (2023); Turki et al. (2023). The queried environment color at $\mathbf{x}$ is denoted as $\mathbf{c}_{\text{env}}$. The final color of the ray is obtained by blending the color of 2DGS projection and the environment map as follows:

$$\mathbf{c}_{\text{env}}(\mathbf{x}) = f(\mathbf{M}, \mathbf{x}) \qquad \mathbf{c}_{\text{final}}(\mathbf{x}) = \mathbf{c}(\mathbf{x}) + (1 - \alpha(\mathbf{x}))\mathbf{c}_{\text{env}}(\mathbf{x}) \qquad (11)$$

where $\mathbf{M}$ denotes the projection matrix from world coordinates to pixel coordinates. $\alpha(\mathbf{x})$ is the rendered alpha map of 2DGS rendering.

**Details of two-stage reconstruction training**

The optimization of our designed 2DGS Huang et al. (2024a) reconstruction for street scenes contains two stages. (1) In the first stage, we employ adaptive density control of 3DGS Kerbl et al. (2023). And $\mathcal{L}_{\text{d}}$, $\mathcal{L}_{\text{n}}$ and $\mathcal{L}_{\text{ds}}$ will be deactivated to reach a more stable initialization of 2DGS reconstruction. (2) In the second stage, $\mathcal{L}_{\text{d}}$, $\mathcal{L}_{\text{n}}$ and $\mathcal{L}_{\text{ds}}$ is activated. As empirically, most 2D Gaussians' opacity will be larger than $0.9$ or lower than $0.1$. The noisy 2DGS with the wrong semantic label will be optimized as low opacity through $\mathcal{L}_{\text{ds}}$. We prune the Gaussians with opacity lower than a threshold $\epsilon$ to further eliminate the noisy semantics in the 3D world, with $\epsilon$ set as $0.3$ in our experiments.

## A.2    DETAILS OF TIME-REVERSAL INPAINTING FRAMEWORK

As is mentioned in Wang et al. (2024b), when we are using a latent-diffusion-based inpainting model, there will be non-ignorable shifts in low-frequency fields if we use images decoded by KL-VAE Kingma & Welling (2022); Rombach et al. (2022) repeatedly for different times. Given that our method can be summarised as inpainting frame $T_i$ with $T_{i+1}$ as a reference through LeftRefill Cao et al. (2024), which is latent-diffusion-based. For a whole sequence of video, if we simply iteratively inpaint every $T_i$ with $T_{i+1}$ as a reference, the shifts in low-frequency fields will be badly augmented by KL-VAE, which will severely harm the quality of our 2D inpainting guidance. To alleviate this inevitable shift from the KL-VAE of the latent diffusion model Rombach et al. (2022). We first select some keyframes in the video. Then we use time-reversal inpainting to inpaint the selected keyframes iteratively in the reversed time sequence.

We firstly select the keyframes of timestamps $\{T_{k_1}, \ldots, T_{k_n}\}$, and we start to inpainting all the keyframes in the reversed time sequence. After we inpaint the keyframe $T_{k_i}$, we generate the middle frames between keyframe $T_{k_{i+1}}$ and keyframe $T_{k_i}$ with keyframe $T_{k_i}$ as reference image. Per-image processing follows Fig. 4. Finally, we will use these results as pseudo-labeled data to further re-optimize the 2DGS of the empty street scene.

To achieve more precise scene optimization, we first identify the 2DGS for removal using hard semantic labels. After removing these 2DGS, we restrict the re-optimization stage to only update Gaussians that are spatially close to the removed ones, ensuring targeted refinement.

# B    MORE EXPERIMENTS

## B.1    ABLATION OF TIME-FORWARD INPAINTING AND TIME-REVERSAL INPAINTING

To further validate the effectiveness of time-reversal inpainting, we do an additional ablation in Sec. 6.2 with time-forward inpainting, which is the reverse version of our proposed time-reversal

inpainting. In Tab. 3 and Tab. 2, our time-reversal inpainting achieves better quantitative results than time-forward inpainting.

For our time-reversal inpainting, we inpaint frame $T_n$ with $T_{n+1}$ as reference. For time-forward inpainting, frame $T_{n+1}$ is inpainted with frame $T_n$ as reference. The Fig. 8 elaborates the details about the process of these two methods. The qualitative comparison in Fig.9 showcases the high-to-low-resolution nature of time-reversal inpainting, which will enhance the quality of the results.

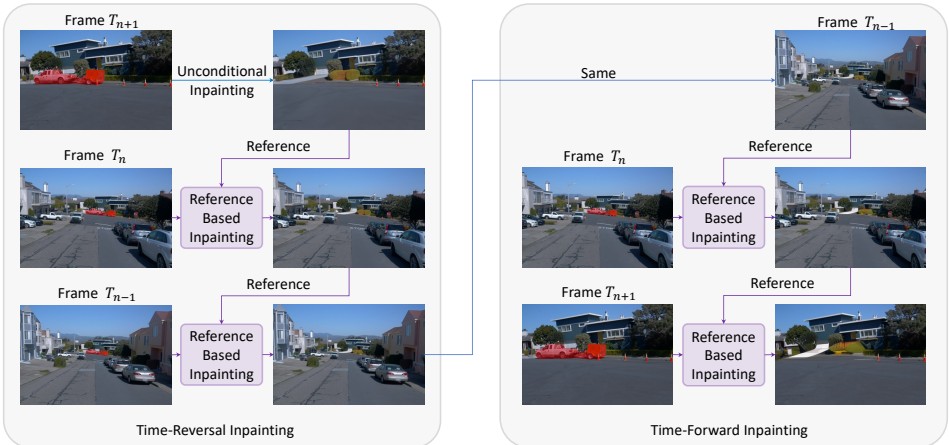

Figure 8: **Illustration of the difference between time-reversal inpainting and time-forward inpainting on inpainting strategy.** We first use unconditional inpainting to inpaint the frame $T_{n+1}$. For our time-reversal inpainting, we inpaint frame $T_n$ with $T_{n+1}$ as reference. For time-forward inpainting, frame $T_{n+1}$ is inpainted with frame $T_n$ as reference.

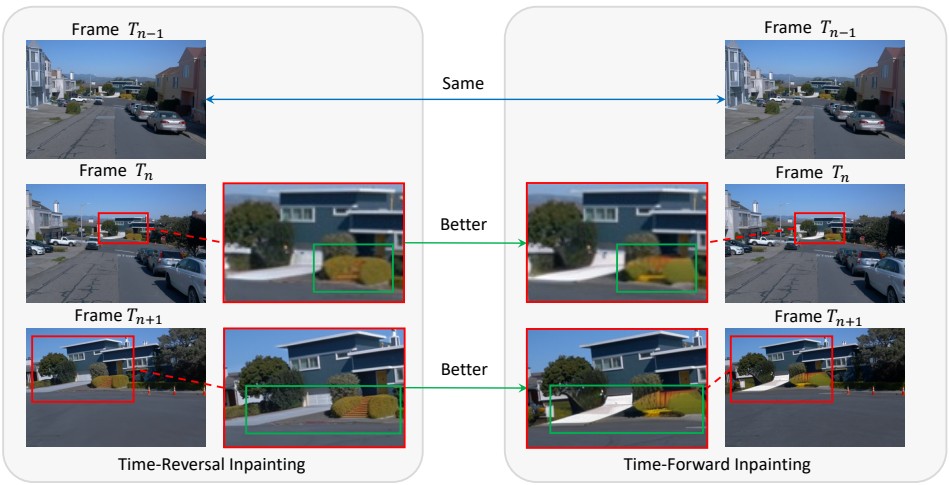

Figure 9: **Illustration of the qualitative comparison between time-reversal inpainting and time-forward inpainting.** We can observe that the quality of time-forward inpainting results degenerates because time-forward inpainting uses a low-to-high-resolution approach. This requires extra super-resolution capacity for the inpainting model to get a better result. However, our time-reversal inpainting uses a high-to-low-resolution approach. High-resolution content will better guide the low-resolution content.

## B.2 ABLATION OF HARD SEMANTIC LABEL

We additionally ablate the effectiveness of the hard semantic label. From Fig. 11, we can observe that both 2DGS representation and hard semantic label contribute to a more stable reconstruction of the semantic field.

The comparison between (a) and (b) demonstrates that the use of hard semantic labels effectively reduces noise within the semantic fields. In addition, the comparison between (a) and (c) indicates

| | Waymo | | Pandaset | |
|---|---|---|---|---|
| | LPIPS↓ | FID ↓ | LPIPS↓ | FID ↓ |
| Time-Forward Inpainting | 0.220 | 136.858 | 0.270 | 158.166 |
| Time-Reversal Inpainting(Ours) | **0.216** | **127.581** | **0.261** | **155.527** |

Table 3: Quantitative ablation of time-reversal inpainting and time-forward inpainting. The result validates the effectiveness of our method.

| | Waymo | | Pandaset | |
|---|---|---|---|---|
| | LPIPS↓ | FID ↓ | LPIPS↓ | FID ↓ |
| LeftRefill Cao et al. (2024) | 0.227 | 135.421 | 0.288 | 168.112 |
| Ours | **0.216** | **127.581** | **0.261** | **155.527** |

Table 4: Quantitative comparison with LeftRefill Cao et al. (2024). The result validates the effectiveness of our method.

that the 2DGS representation leads to more stable semantic fields. Finally, (d) illustrates the clean and stable semantic field achieved by employing hard-label semantic 2DGS in our method.

By reconstructing a clean and stable semantic field of the street scene, we can more accurately identify the Gaussians that need to be removed. This facilitates obtaining a high-quality 2D inpainting result, which serves as effective guidance for re-optimization.

### B.3 COMPARISON WITH LEFTREFILL

We additionally discuss the qualitative comparison with LeftRefill Cao et al. (2024) as another baseline. Since LeftRefill requires an image as a reference, LeftRefill can't be naturally run as unconditional inpainting methods like LAMA Suvorov et al. (2021) and SDXL Podell et al. (2023) in Tab. 1. We adapt LeftRefill with the 10th future frame as a condition and use the mask obtained after our reconstruction stage. LeftRefill is also operated in a reverse order.

From Tab. 4, we can observe that our time-reversal inpainting pipeline generates better results than LeftRefill. From Fig. 10, we observe that due to this adapted naive reverse inpainting with LeftRefill taking the future frame as a reference, some regions are not visible in the future frame. This limitation can lead to low-quality inpainting, as highlighted in the red frame of LeftRefill's result. In contrast, our pipeline generates a more natural inpainting result through 2DGS re-optimization, ultimately achieving a clear 3D inpainting result.

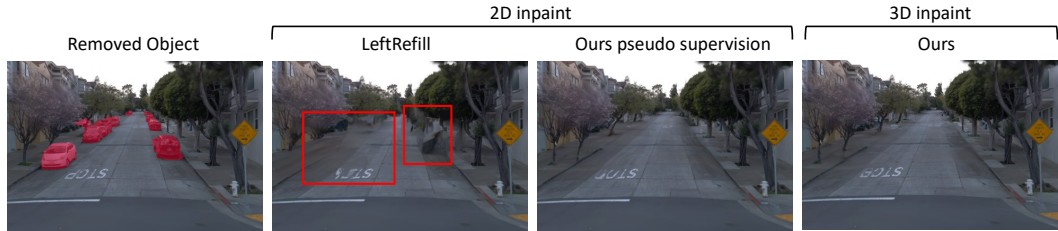

Figure 10: Illustration of comparison with LeftRefill Cao et al. (2024). Since naive reverse inpainting with LeftRefill takes the future frame as a reference, some regions are not visible in the future frame. We observe that this will lead to a low quality of inpainting, as highlighted in the red frame of LeftRefill's result. Our pipeline generates a more natural inpainting result for 2DGS re-optimization and finally obtains a clear 3D inpainting result.

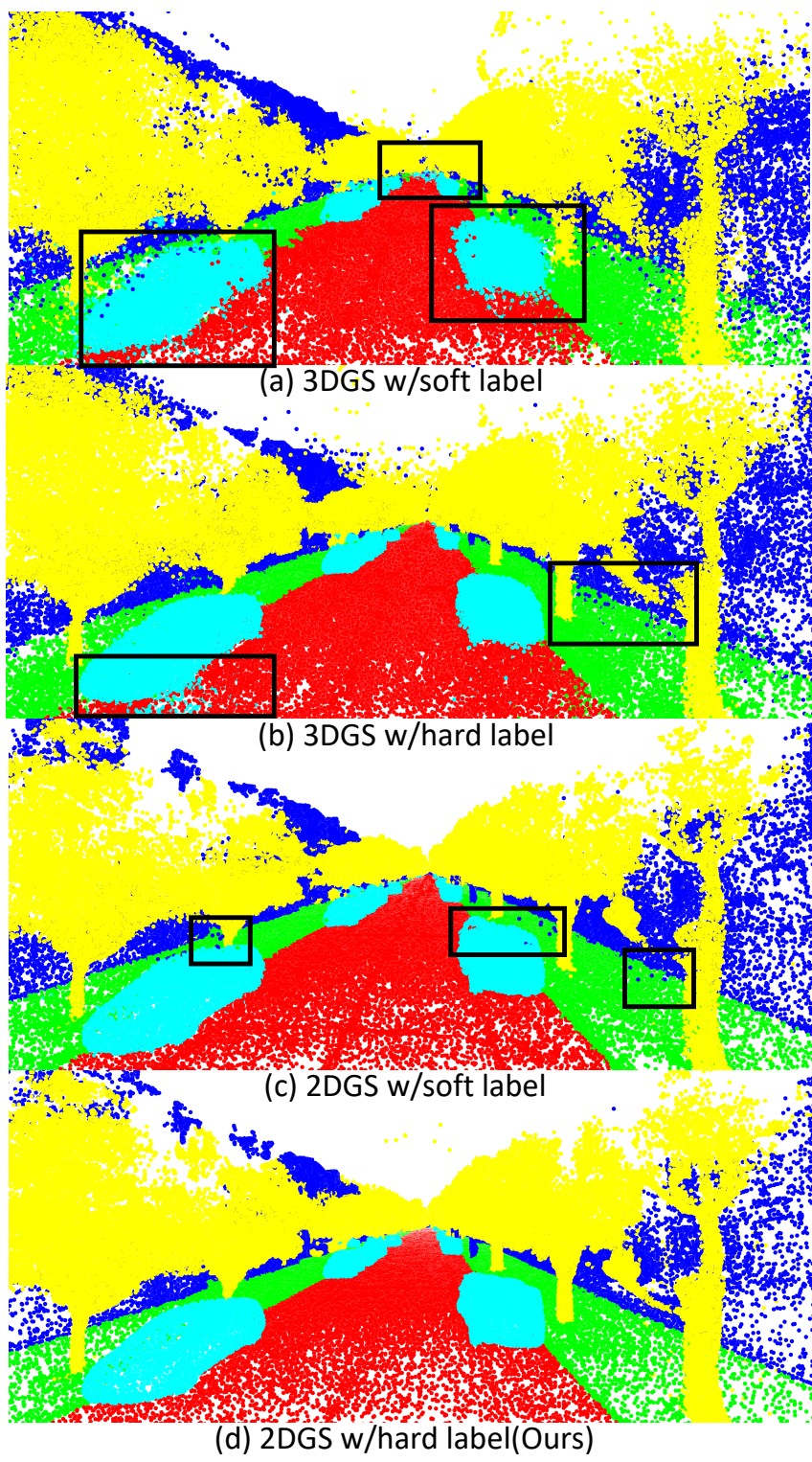

Figure 11: Illustration of hard semantic label ablation. Black rectangles in the figures include the noise in the semantic fields. The comparison between (a) and (b) demonstrates that hard semantic labels effectively reduce noise within semantic fields. Similarly, the comparison between (a) and (c) indicates that the 2DGS representation contributes to more stable semantic fields. Finally, (d) illustrates the clean and stable semantic field achieved by employing hard-label semantic 2DGS in our method.

| Dataset(Frames) | Reconstruction | Our Inpaint | Re-optimize |
|---|---|---|---|
| Waymo(198) | 10016 sec | 1035 sec | 524 sec |
| Pandaset(80) | 9640 sec | 311 sec | 257 sec |

Table 5: Computational cost of each stage with our pipeline(in seconds): We evaluate on both Waymo and Pandaset. We use 8 scenes from each dataset and average their time consumption in experiments. "Reconstruction" is the main efficiency bottleneck in our pipeline.

| Method | Waymo (198) | Pandaset (80) |
|---|---|---|
| LaMa Suvorov et al. (2021) | 0.20 sec | 0.28 sec |
| SDXL Podell et al. (2023) | 5.18 sec | 5.21 sec |
| Propainter Zhou et al. (2023a) | 0.59 sec | 0.53 sec |
| Ours | 5.22 sec | 5.09 sec |

Table 6: Total per-frame time cost comparison from 2D inpainting aspect: We use 8 scenes from each dataset and average their time consumption in experiments. Our method is comparable to SDXL for per-frame time cost.

### B.4 QUALITATIVE COMPARISON OF RENDERED GEOMETRY

Since we want to reconstruct the empty street, we also want to compare the geometry property of our method other than just appearance. From Fig. 14, Fig. 15, Fig. 16, Fig. 17, we can observe that our method produces both better appearance quality and geometry quality from rendered RGB and rendered normal images.

### B.5 COMPUTATIONAL ANALYSIS

We additionally conduct a computational analysis of each stage in our pipeline and per-frame inpainting time cost. From Tab. 5, we observe that the "Reconstruction" stage is the main efficiency bottleneck in our pipeline. By utilizing recent techniques like Instantsplat Fan et al. (2024), our whole pipeline has the potential to be accelerated and may reach the result of reconstructing an empty street in 30 minutes. From Tab. 6, our pipeline isn't inferior to the Diffusion-based Podell et al. (2023) method from the efficiency aspect.

## C ADDITIONAL RESULTS

### C.1 EMPTY STREET SCENE MESH EXTRACTION

We can further extract the mesh for our reconstructed empty street scene using TSDF fusion following 2DGS Huang et al. (2024a) with Open3D Zhou et al. (2018). In Fig. 19 and Fig. 20, we visualize the extracted colored mesh before and after our unveiling. Our inpainting framework can successfully remove unwanted cars from the street and finally reconstruct an empty street in mesh representation. Mesh extraction results further verify the correct geometry produced through our method.

We clarify that our problem formulation is not exactly the same as StreetSurf Guo et al. (2023). The target of our method is to reconstruct the empty street. However, the extracted mesh and reconstructed street of StreetSurf is the original static scene "before unveiled".

Another key difference is that our setting lacks ground-truth "after unveiled" training data for both Lidar and images. StreetSurf relies on ground-truth "before unveiled" data for both training and evaluation.

While we are still able to evaluate the reconstructed "before unveiled" scenes to compare with StreetSurf, which will provide meaningful insights for future work.

Following StreetSurf, we utilize the Lidar data under the real-world scale for geometry evaluation. In StreetSurf, the extracted mesh may include some parts outside of the scene. To ensure a fair

| Method | CD↓ | F-Score↑ |
|---|---|---|
| StreetSurf Guo et al. (2023) | **0.52** | 56.70 |
| Ours(before unveiled) | 0.55 | **61.54** |

| Geometry-related input | Lidar | Monocular Prior |
|---|---|---|
| StreetSurf Guo et al. (2023) | ✓ | ✓ |
| Ours(before unveiled) | ✓ | |

Table 7: Comparison of the reconstruction performance on "before unveiled" geometry with Street-Surf Guo et al. (2023) on 24 scenes from Waymo dataset Sun et al. (2020). **Left:** We observe that our Chamfer Distance is higher than StreetSurf, while F-Score is higher than StreetSurf. The reconstruction performance of observed geometry appears to be comparable. **Right:** The checkbox of the geometry-related input data. We run StreetSurf with both monocular prior and Lidar as input. While in our approach, we operate without relying on the monocular geometry prior.

comparison, we crop these out-of-range meshes from the extracted mesh. Specifically, any mesh that is more than 5 meters away from the closest Lidar point will be cropped.

For the experimental setup, we select both Chamfer Distance(CD) and F-score with a 0.25-meter threshold as our geometry evaluation metrics. We evaluate StreetSurf using both monocular geometry priors and Lidar as geometry-related inputs, while our method is tested exclusively with Lidar as the geometry-related input.

From Tab. 7, we observe that our Chamfer Distance is greater than that of StreetSurf, while our F-Score surpasses StreetSurf's. From Fig. 12, we empirically observe that the StreetSurf achieves higher accuracy on the observable ground of the street. Even after fair mesh cropping, the reconstructed mesh of StreetSurf is still affected by out-of-range meshes to some extent. This results from the inherent methodological differences between SDF extraction and TSDF fusion, which may lead to a slightly lower F-Score for the SDF extraction approach. Overall, the reconstruction performance of the observed geometry appears to be comparable.

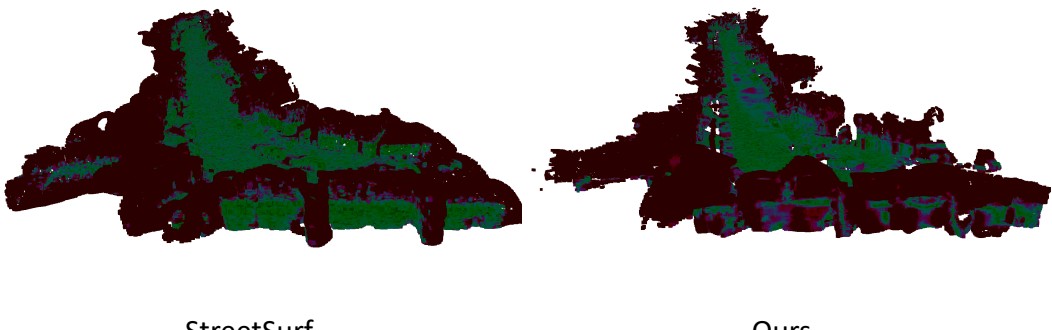

StreetSurf        Ours

Figure 12: Illustration of the "before unveiled" geometry comparison with StreetSurf Guo et al. (2023). "Green" indicates "more accurate" regions, while "Red" represents "less accurate" regions. We empirically observe that the StreetSurf achieves higher accuracy on the observable ground of the street. Even after fair mesh cropping, the reconstructed mesh of StreetSurf is still influenced by out-of-range meshes to some extent. This effect stems from inherent methodological differences between SDF extraction and TSDF fusion, which may contribute to a slightly lower F-Score for the SDF extraction approach. Overall, the reconstruction performances are comparable.

### C.2 EXAMPLE OF REMOVING THE STANDING PEDESTRIAN

As in Fig. 21, we highlight an example of removing the standing pedestrian from the scene.

### C.3 MORE VISUAL COMPARISON

We provide additional qualitative comparisons of inpainting results for the Pandaset dataset Xiao et al. (2021) in Fig. 24.

### C.4 Video visualizations

In order to conveniently view our video results, we prepare a web viewer at "./index.html" from the root path of the supplementary materials.

#### C.4.1 Novel view synthesis videos visualizations

As in Fig. 23, we showcase two novel view synthesis videos. The file paths are illustrated.

#### C.4.2 More video visualizations

As in Fig. 25, we visualize three scenes involved in Tab.1 for better comparison. The file paths are illustrated. From video comparison, it can be observed that our method outperforms other baselines.

## D Discussion on semantic label supervision

We use SegFormer Xie et al. (2021) as the pre-trained model for segmentation, and our results appear to be robust as a whole. Both our quantitative and qualitative results show that our method is stable with SegFormer.

While it is inevitable that certain failure cases occur with small objects or object corners, these challenges are common across most segmentation methods. The example cases are illustrated in Fig. 13. As segmentation techniques continue to evolve, our method is poised to benefit and improve alongside them.

## E Discussion on dynamic object removal

We illustrate a case to handle simple dynamic object removal in Fig. 22, which can be observed in the scene 3(a moving car) in the supplementary videos. For more challenging dynamic cases, utilizing optical flow or dynamic object detection over the video sequence to identify dynamic objects from 2D may be a considerable solution.

Alternatively, we can model dynamic Gaussians in street scenes like Chen et al. (2023b) and Yan et al. (2024) for more challenging dynamic object removal.

The separation of dynamic and static elements may also facilitate the removal of dynamic objects from scenes. Current methods, such as StreetGaussians Yan et al. (2024) (which utilizes 3D box-based scene decomposition) and S3Gaussian Huang et al. (2024b) (a self-supervised approach to scene decomposition), aim to distinguish between dynamic and static components. However, these methods may not consistently differentiate static removable objects (like stopping cars) from essential scene elements (like traffic signs).

## F Discussion on 360° or wide range surrounding videos

Our time-reversal inpainting pipeline works for "forward-facing" cameras. Thus, we used frontal cameras for the experiment, which is the same as the experimental setup in Yan et al. (2024); Chen et al. (2023b); Zhou et al. (2024). We found the inpainting results of the frontal view sufficient for recovering a 3D unveiled street with satisfactory geometry properties. For left-back and right-back(side and back cameras), our method doesn't naturally promise to maintain the consistency of the inpainting. The homography technique may help maintain consistency between different views by leveraging the overlaps in images across those views.

## G Societal impacts

This technology can distort public space representations in urban planning, potentially leading to flawed decisions. Additionally, it may be misused to alter important archaeological sites in digital reconstructions, resulting in misinformation about historical facts.

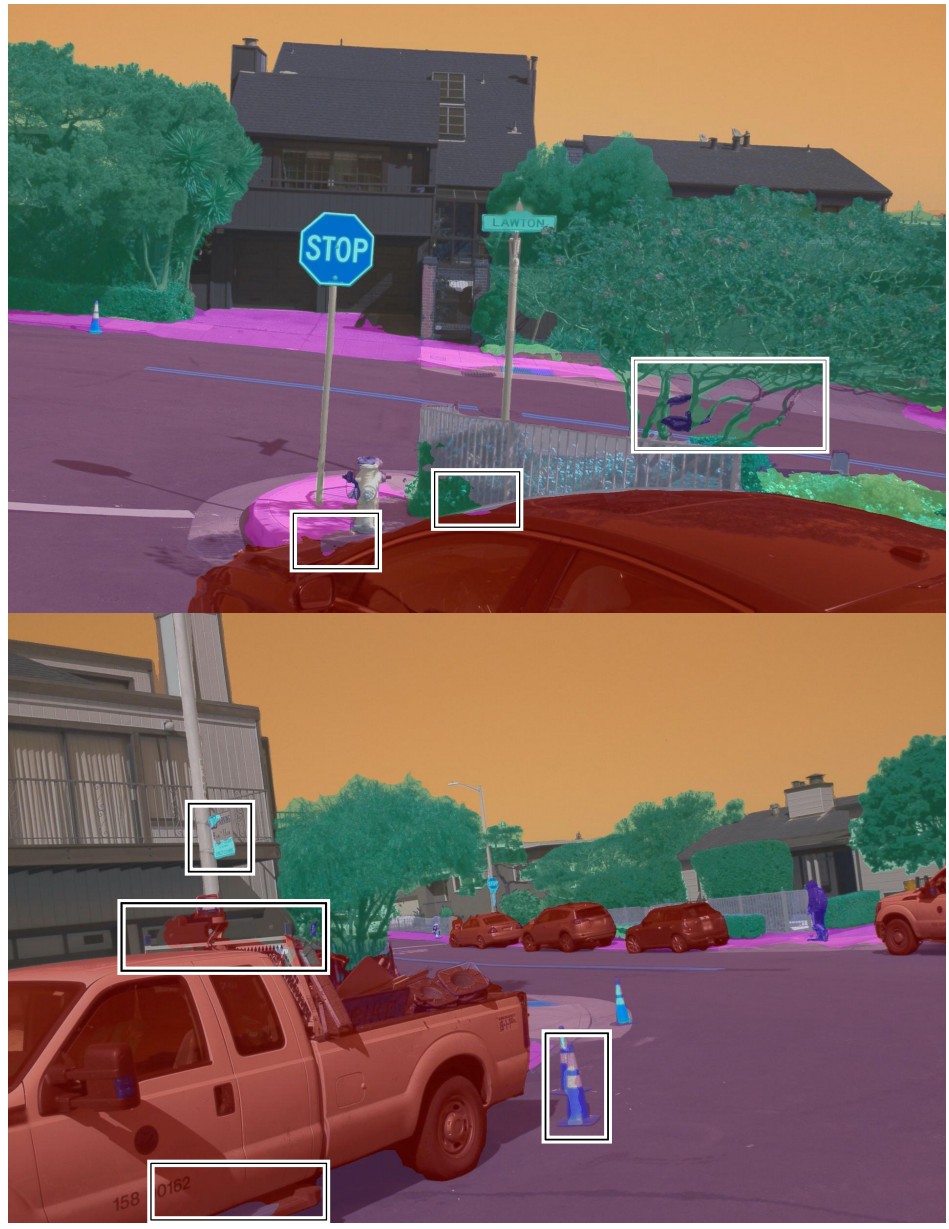

Figure 13: For the semantic segmentation mask predicted by SegFormer Xie et al. (2021) in original full resolution, some noise exists at the boundaries between regions with different semantic tags. These failure cases are more likely to occur at the corners of small objects.

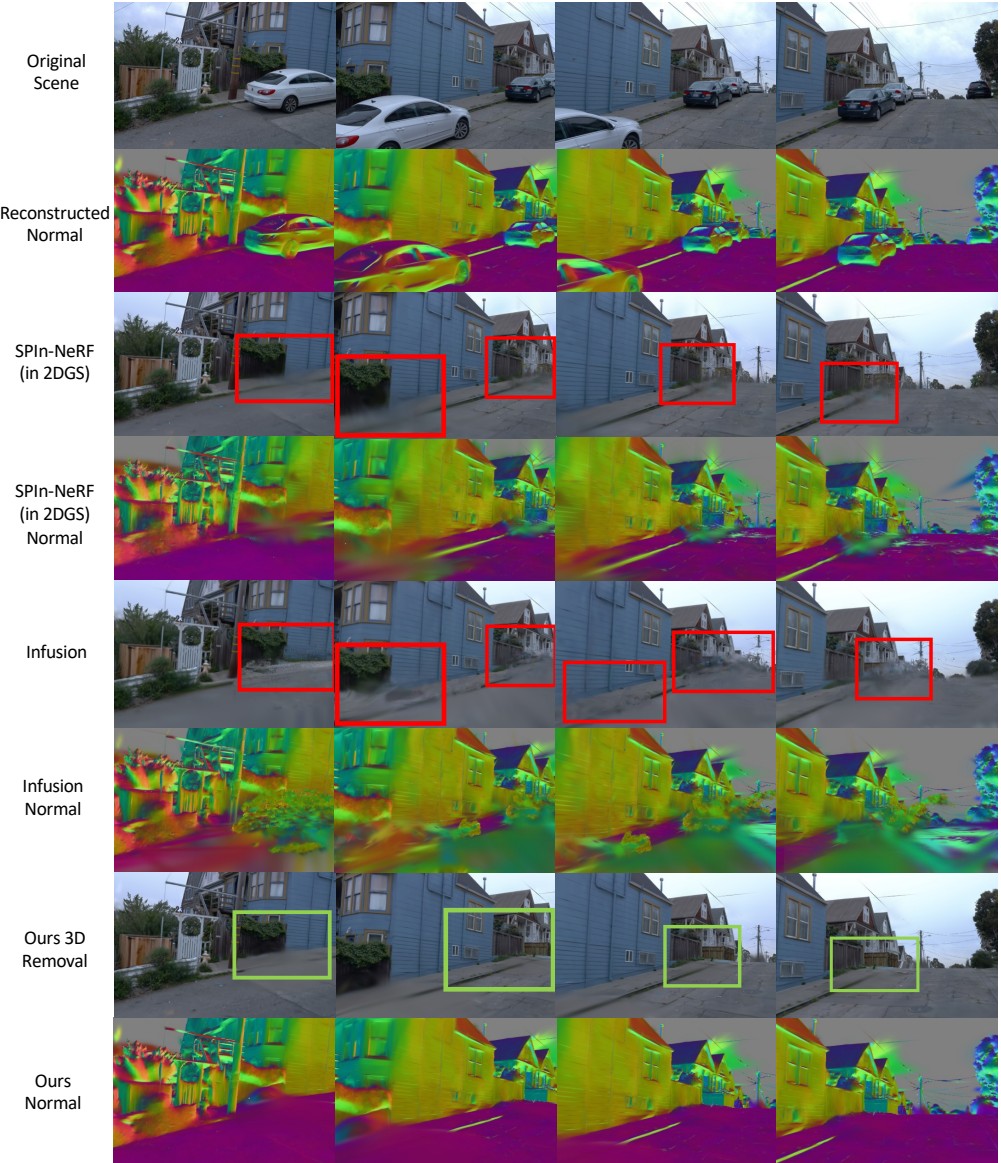

Figure 14: Illustration of geometry performance comparison.

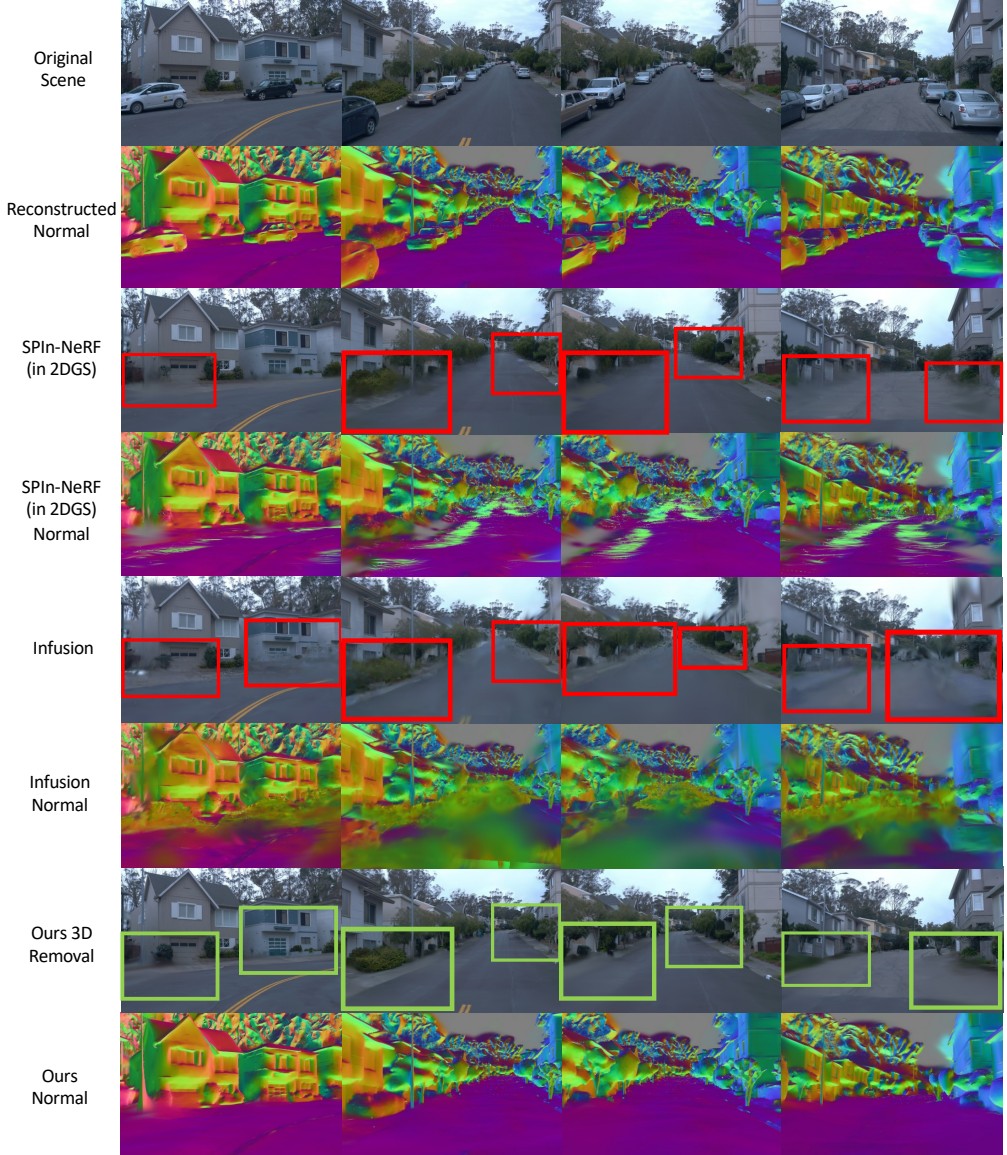

Figure 15: Illustration of geometry performance comparison.

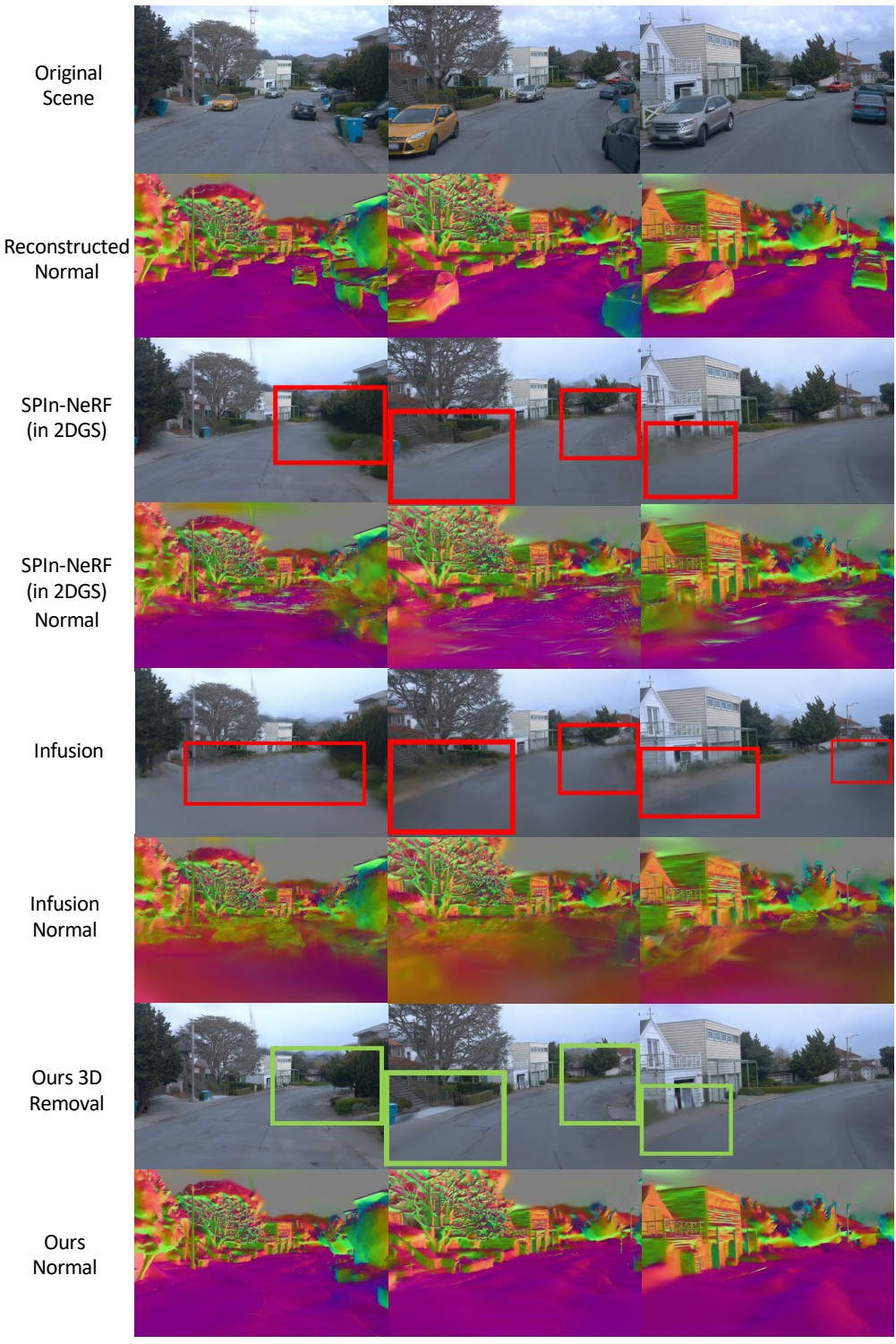

Figure 16: Illustration of geometry performance comparison.

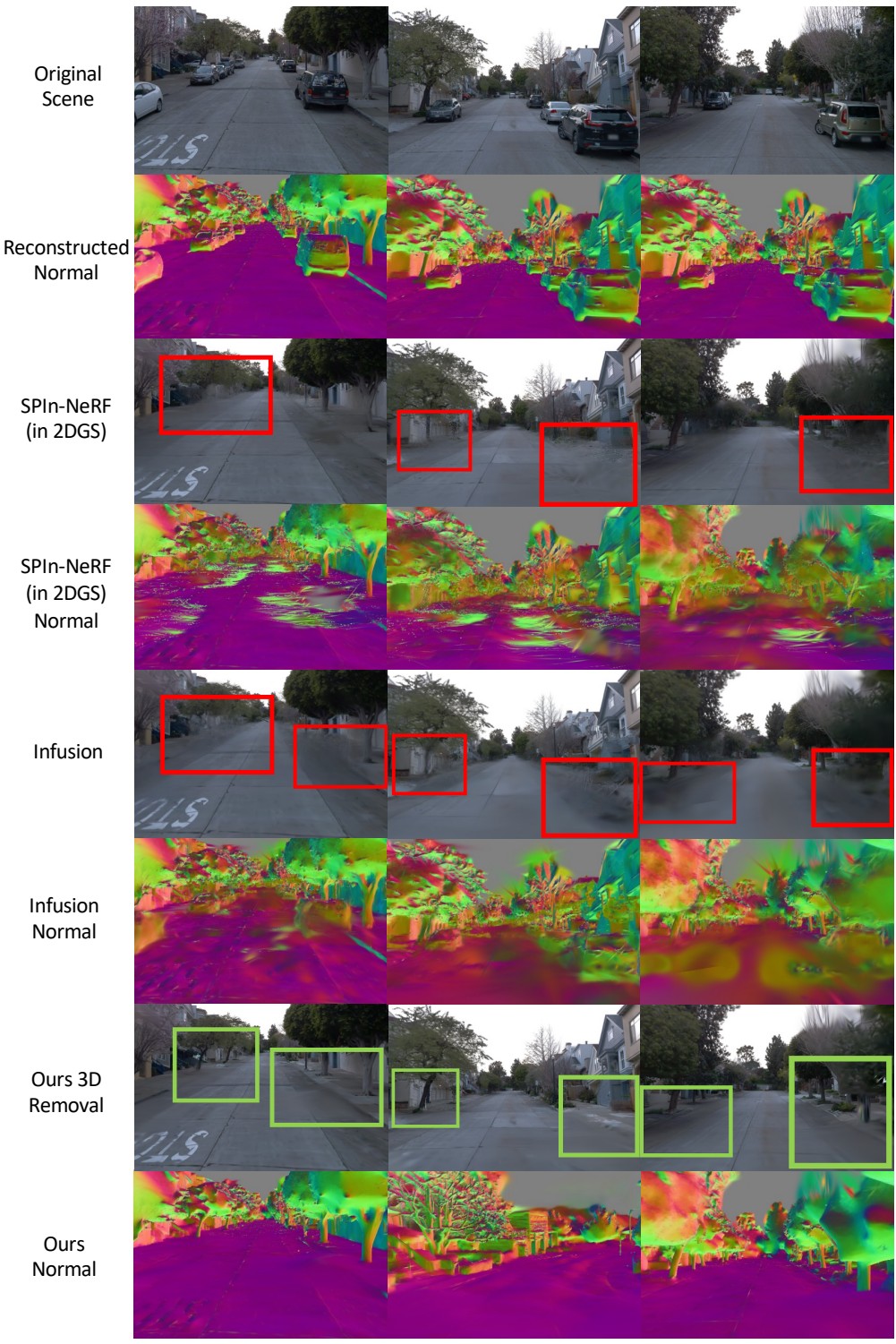

Figure 17: Illustration of geometry performance comparison.

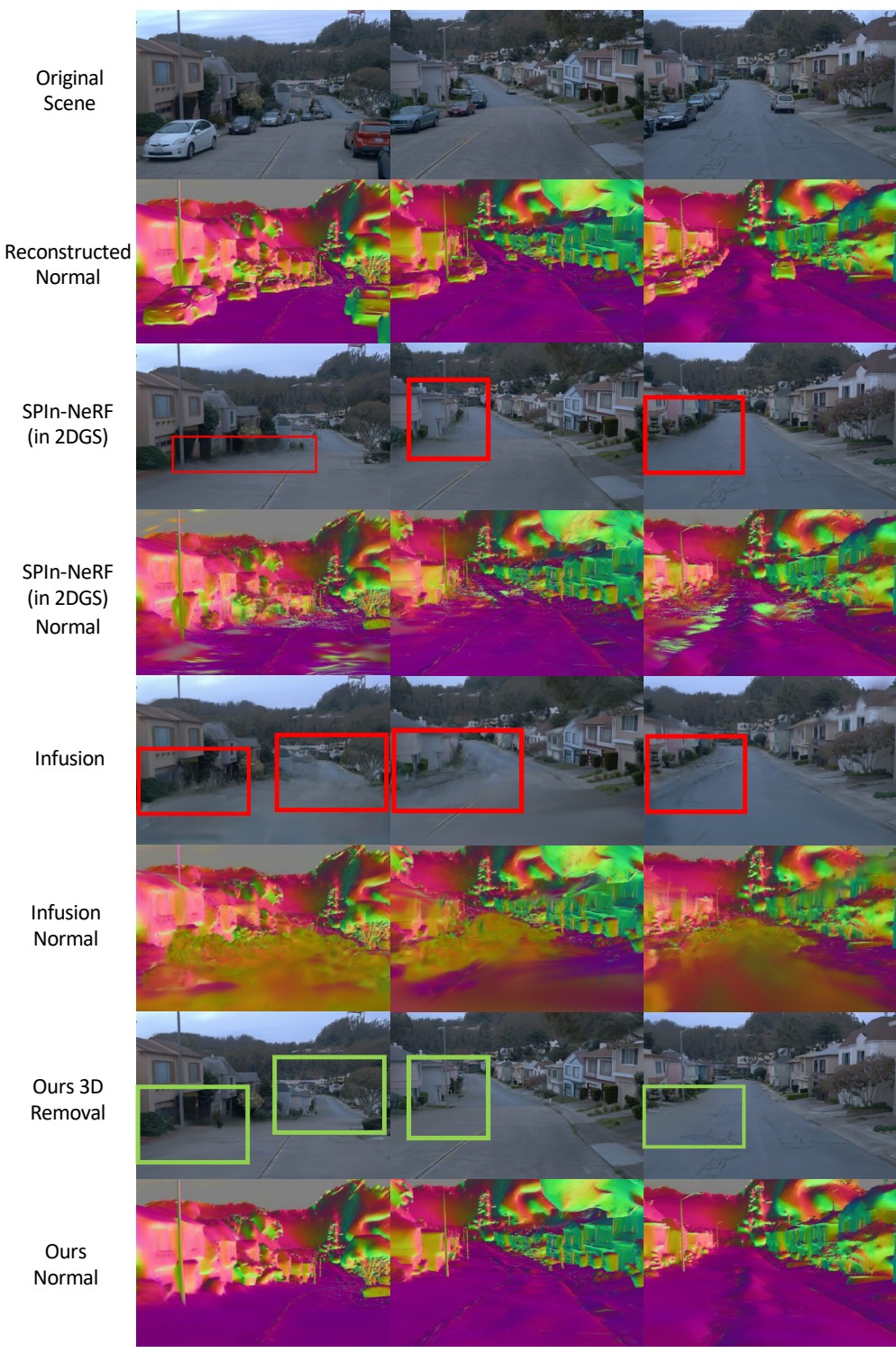

Figure 18: Illustration of geometry performance comparison.

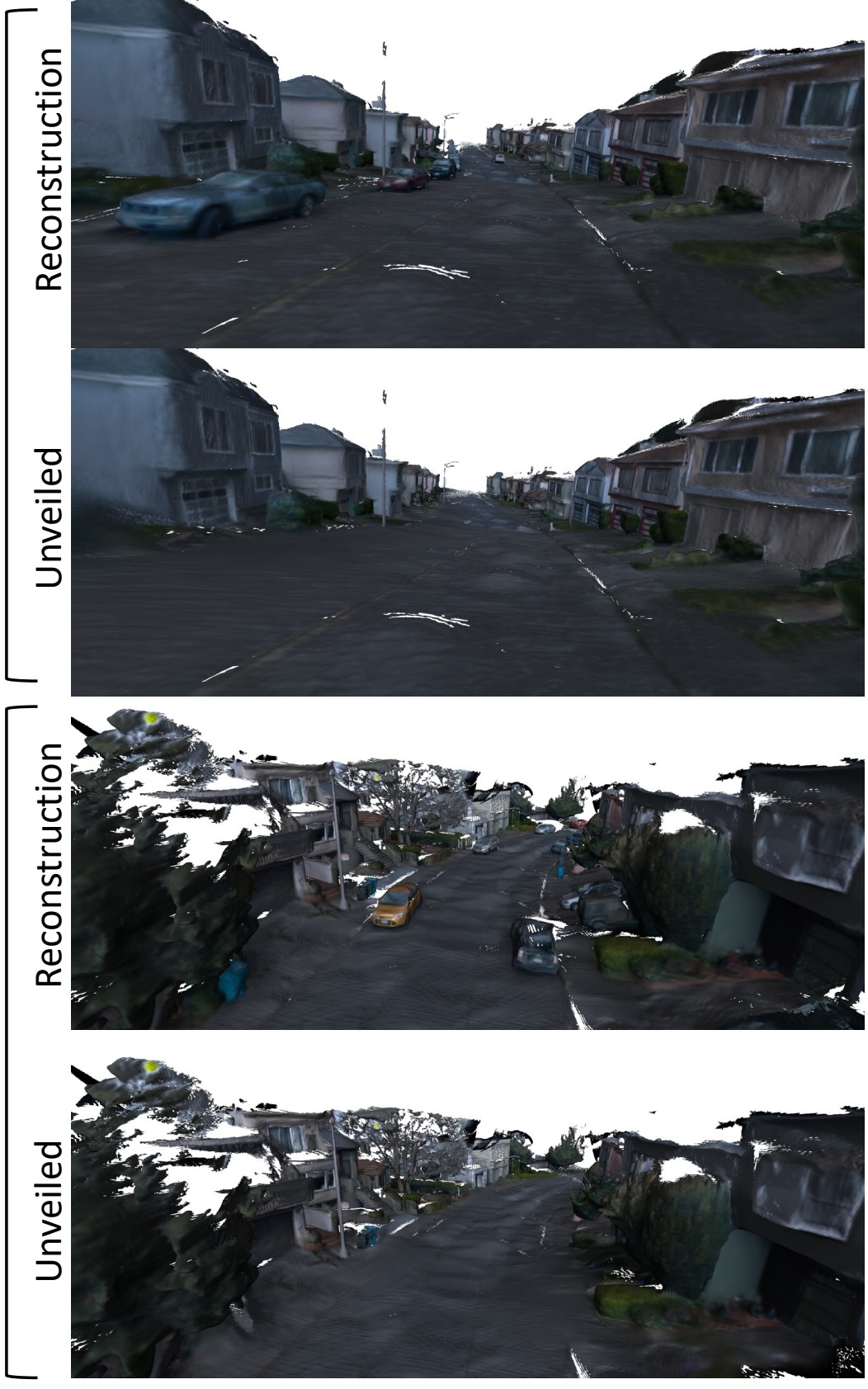

Figure 19: Illustration of colored mesh comparison between before and after the unveiling.

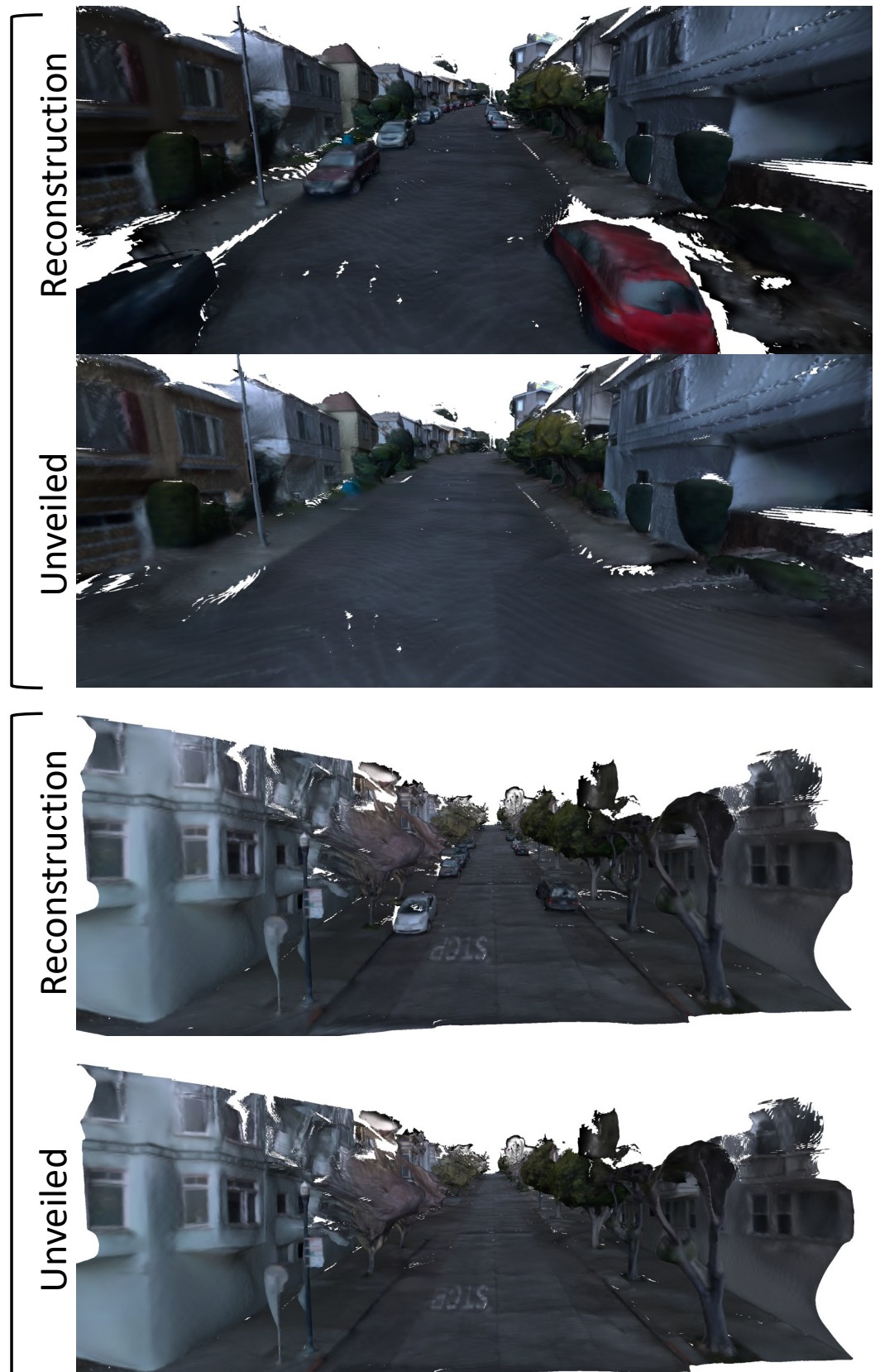

Figure 20: Illustration of colored mesh comparison between before and after the unveiling.

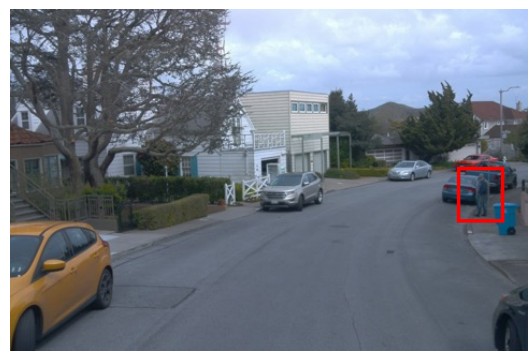 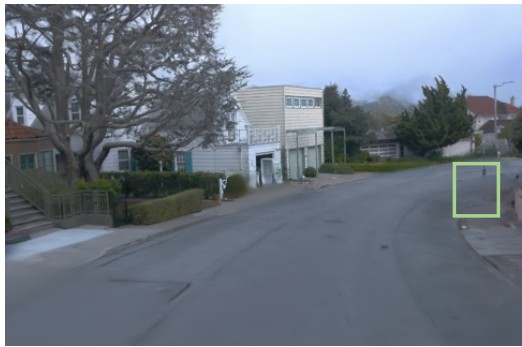

Training data          Our unveiled

Figure 21: Illustration of removing the standing pedestrian.

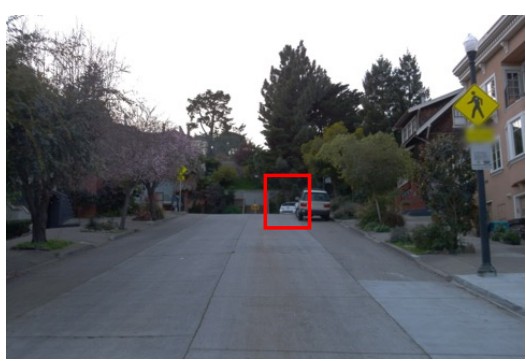 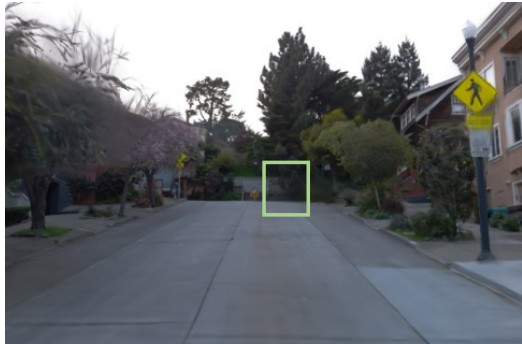

Training data          Our unveiled

Figure 22: Illustration of removing simple dynamic case.

# Novel View Synthesis Videos

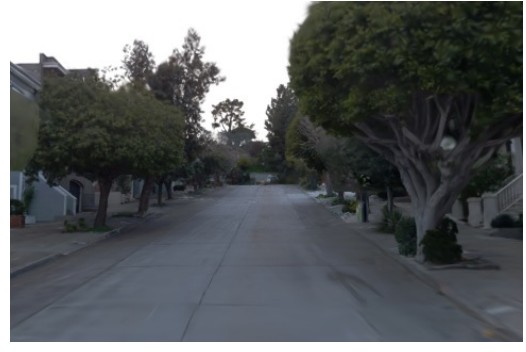 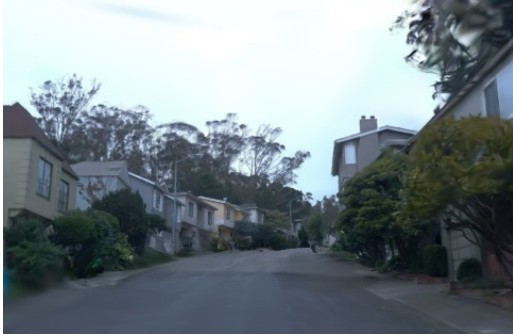

./static/videos/NVS/nvs1.mp4      ./static/videos/NVS/nvs2.mp4

Figure 23: Illustration of novel view synthesis videos and their file paths. **It's recommended to open our web viewer located at "./index.html"**.

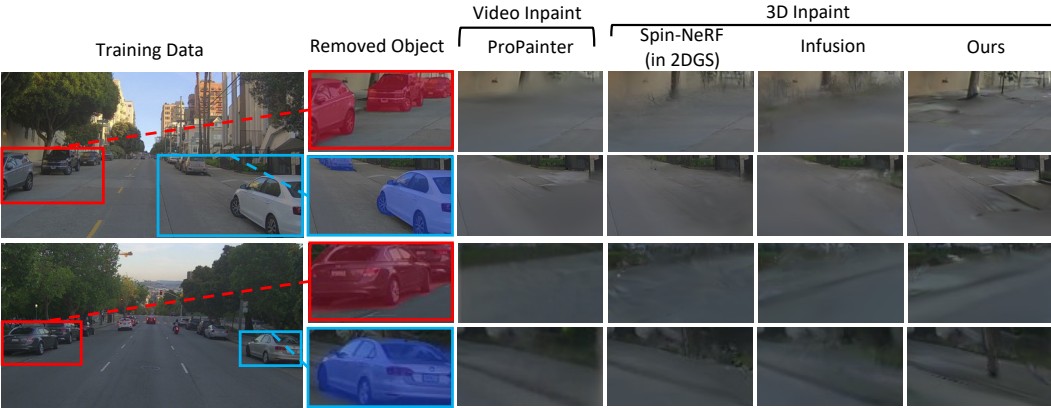

Figure 24: More qualitative comparison on Pandaset dataset Xiao et al. (2021). Our method produced a clearer result of the ground and trees behind the removed object compared to the baselines.

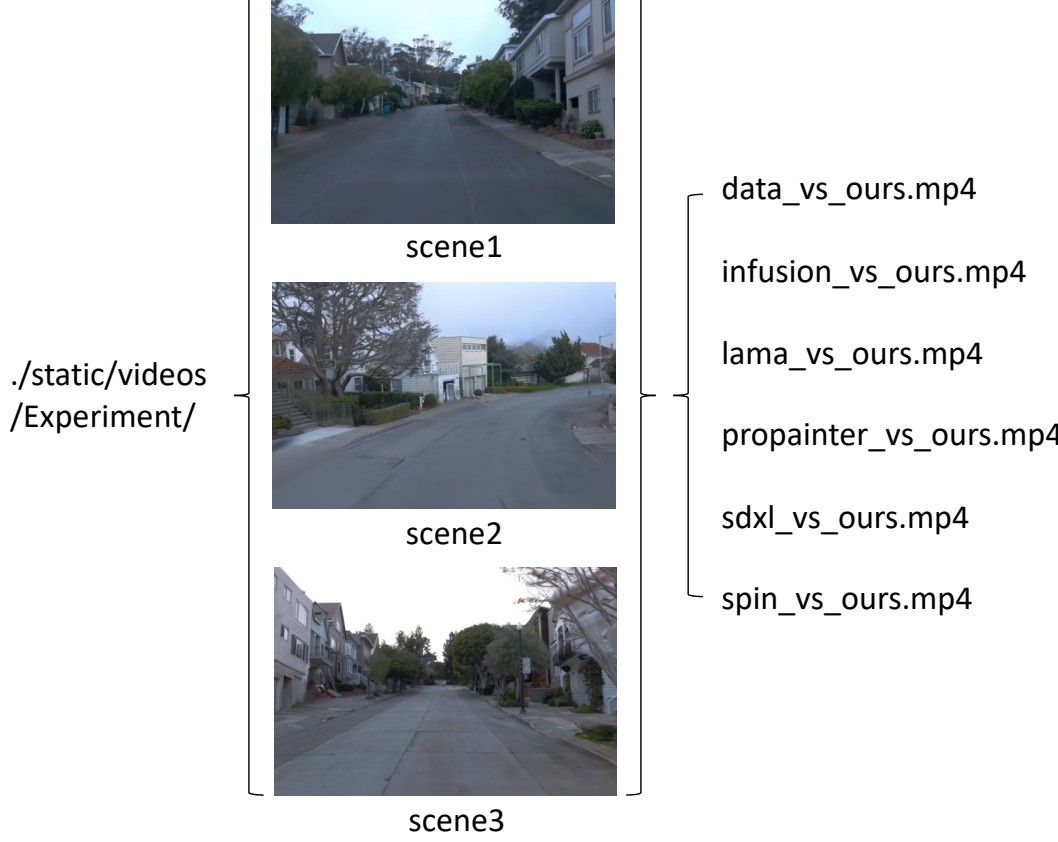

Figure 25: Illustration of video comparison with baseline and their file paths. **It's recommended to open our web viewer located at "./index.html"**. "data_vs_ours.mp4" shows our results compared with training data for visualization. "infusion_vs_ours.mp4" shows both RGB and normal results between Infusion and our method. "lama_vs_ours.mp4" shows our results compared to LaMa. "propainter_vs_ours.mp4" shows our results compared to ProPainter, which is a state-of-the-art video inpainting method. "sdxl_vs_ours.mp4" shows our results compared to SDXL. "spin_vs_ours.mp4" shows both RGB and normal results between SPIn-NeRF in 2DGS representation and our method.

