# OpenReview forum: "3D StreetUnveiler with Semantic-aware 2DGS - a simple baseline"
_ICLR.cc/2025/Conference — ICLR 2025 Poster_

### Official Review · Reviewer_SLKs · 2024-10-31

**Soundness:** 4
**Presentation:** 4
**Contribution:** 4
**Rating:** 8
**Confidence:** 5

**Summary:**

The paper presents StreetUnveiler, a method based on semantic 2DGS to achieve the empty street scene reconstruction.
StreetUnveiler first utilizes the rendered alpha map to locate unobservable regions with the aid of distortion loss and shrinking loss.
Then a time-reversal inpainting framework is introduced to optimize temporal consistency by leveraging later frames as references for earlier ones.
Experiments are conducted on Waymo and Pandaset to demonstrate the effectiveness of StreetUnveiler for the empty street reconstruction.

**Strengths:**

1. The writing of the paper is clear and easy to follow.
2. The task of empty street scene reconstruction is interesting. The limitations discussed in the introduction are well studied in the following section.
3. The proposed method is evaluated on two datasets. Many quantitative and qualitative experiments are conducted to demonstrate the effectiveness of this method.

**Weaknesses:**

1. Lack of Quantitative Metrics for Mesh: While the authors discuss the differences in problem formulation between StreetSurf and StreetUnveiler, a quantitative comparison of mesh reconstruction is still lacking. StreetSurf uses point clouds as ground truth to calculate Chamfer Distance as a metric. Could a point cloud derived from Structure from Motion (SfM) be used as a ground truth for evaluating StreetUnveiler’s performance? Even relative values across different baselines could provide meaningful insight.
2. Limitations in Semantic Label Supervision: Current semantic supervision seems limited to common objects like vehicles, yet street environments also include pedestrians and less common obstacles. Could expanding the range of semantic (instance) labels improve the model’s robustness across diverse urban scenes?

**Questions:**

1. Applications of Unveiled Street Scene Reconstruction: The task of reconstructing an unveiled street scene is indeed challenging and engaging. However, real-world street environments include more than just vehicles, and vehicles can often obscure parts of other objects, such as the lower half of a road sign. Would it be more practical to first apply 2DGS to separate scenes based on specific requirements, then proceed with targeted inpainting? This approach could provide more utility by selectively reconstructing essential scene elements.
2. Potential to Build on Existing 3DGS Street Scene Reconstruction Methods: Current methods, such as Street Gaussians[1] (using 3D box-based scene decomposition) and S3Gaussian[2] (self-supervised scene decomposition), aim to separate scenes into dynamic and static components. Could integrating these approaches enhance StreetUnveiler’s ability to more effectively distinguish between removable objects and essential scene elements? This could potentially improve the precision in identifying and preserving critical components.

[1]Yan Y, Lin H, Zhou C, et al. Street gaussians for modeling dynamic urban scenes[J]. arXiv preprint arXiv:2401.01339, 2024.
[2]Huang N, Wei X, Zheng W, et al. $\textit {S}^ 3$ Gaussian: Self-Supervised Street Gaussians for Autonomous Driving[J]. arXiv preprint arXiv:2405.20323, 2024.

---

> ### Author Response · Authors · 2024-11-18
>
> We respectfully acknowledge the reviewer SLKs’s valuable comments on our work to improve this paper. We’d be delighted if your concerns could be solved by our reply.
>
> Reply to Strengths:
>
> We appreciate the reviewer recognition of our new setting and various experiments.
>
> Reply to Weaknesses & Questions:
>
> 1. “While the authors discuss the differences in problem formulation between StreetSurf and StreetUnveiler, a quantitative comparison of mesh reconstruction is still lacking. StreetSurf uses point clouds as ground truth to calculate Chamfer Distance as a metric. Could a point cloud derived from Structure from Motion (SfM) be used as a ground truth for evaluating StreetUnveiler’s performance? Even relative values across different baselines could provide meaningful insight.”
>
> We really appreciate the reviewer’s concern. This is really helpful to the enhancement of our paper. As mentioned in Section.A.1 of our supplementary PDF. Both SfM points and Lidar points are both used in our reconstruction stage. These points all model the original “before unveiled” scene. Here we call them “before unveiled points”(BUPoints) for simplicity.
>
> The reconstructed mesh of StreetSurf should be strictly aligned with BUPoints. The StreetSurf doesn’t recover the unobserved geometry. In contrast, our “time-reversal inpainting” based StreetUnveiler wishes to recover the unobserved geometry occluded by the removed objects. In this case, since BUPoints don’t include the unobserved geometry, evaluation with Chamfer Distance from BUPoints may not be proper. Consequently, StreetSurf may not be a proper baseline to be compared with, because it doesn’t contain the result of our targeting unobserved geometry.
>
> We respectfully welcome the reviewer to share more existing concerns about this part. We are very willing and eager to solve your concerns further.
>
> 2. “Current semantic supervision seems limited to common objects like vehicles, yet street environments also include pedestrians and less common obstacles. Could expanding the range of semantic (instance) labels improve the model’s robustness across diverse urban scenes?”
>
> We appreciate the reviewer for sharing the concern. We provided an example of removing the pedestrian in Figure.18 of our supplementary PDF.
>
> For the expansion of the range of semantic labels, as long as the semantic prediction from a pretrained semantic segmentation method is correct, a finer granularity of semantics may potentially facilitate the removal with finer granularity. As the dataset we used is an autonomous driving-oriented dataset, vehicles and pedestrians are more common; thus, we mainly remove them from our current method. In the future, it would be an interesting direction to study our method on more diverse datasets.
>
> 3. “However, real-world street environments include more than just vehicles, and vehicles can often obscure parts of other objects, such as the lower half of a road sign. Would it be more practical to first apply 2DGS to separate scenes based on specific requirements, then proceed with targeted inpainting? This approach could provide more utility by selectively reconstructing essential scene elements.”
>
> We appreciate the reviewer’s valuable suggestions. This is a possible solution for a special targeted inpainting. From the technical side, our method can remove other objects on the road,  if a pretrained semantic segmentation model performs well. For example, if we focus on recovering a road sign as an example, we can specially separate the specific vehicle that obscures the sign, and then remove that specific vehicle to proceed with targeted inpainting. To identify this specific vehicle, we can cluster the 2DGS with vehicle semantics into separate clusters. Then, the user will be able to select the specific vehicle cluster that is required for targeted inpainting. As we mentioned, the main issue depends on the performance of the semantic segmentation method.
>
> 4. “Current methods, such as Street Gaussians[1] (using 3D box-based scene decomposition) and S3Gaussian[2] (self-supervised scene decomposition), aim to separate scenes into dynamic and static components. Could integrating these approaches enhance StreetUnveiler’s ability to more effectively distinguish between removable objects and essential scene elements? This could potentially improve the precision in identifying and preserving critical components.”
>
> We appreciate the reviewer’s question. We believe the separation of dynamic and static would help the removal of dynamic objects. However, it may not work all the time to distinguish between removable objects and essential scene elements. Examples would be stopping vehicles in Scene 1 of the supplementary video we provided. They are static in videos, but they are not essential elements in the scene. We will cite [1] and [2] and add a discussion about this to further enhance our paper in our revision.
>
> We thanks again for your valuable questions.

---

> > ### Comment · Reviewer_SLKs · 2024-11-18
> > **Response to Rebuttal**
> >
> > Thank you for your detailed rebuttal. I appreciate your clarifications, but I still have two concerns:
> >
> > 1. While I understand your point that "StreetSurf may not be a proper baseline for comparison as it doesn’t address unobserved geometry," would it be possible to compare the observed geometry between StreetUnveiler and StreetSurf by applying 3D masks to the reconstructed meshes?
> >
> > 2. It appears that StreetUnveiler's performance may rely significantly on the capability of the segmentation model. If the segmentation contains noise, can StreetUnveiler still maintain robust performance under such conditions?

---

> > > ### Author Response · Authors · 2024-11-18
> > >
> > > Dear reviewer,
> > >
> > > We are very willing to address your remaining concerns mentioned in your last reply.
> > >
> > > **Reply to Question 1:**
> > >
> > > The comparison is feasible. We evaluated StreetSurf and our reconstructed scene on the observed geometry using the same 24 scenes from the Waymo dataset as in the main experiment.
> > >
> > > We evaluate the geometry under the real world scale and with the lidar data.
> > >
> > > For StreetSurf, the extracted mesh may contain some meshes outside of the scene. For a fair comparison, we crop the out-of-range meshes from the extracted mesh. (The mesh that is 5m far from the closest lidar point will be cropped.)
> > >
> > > We select both Chamfer Distance and F-score with 0.25m as threshold.
> > >
> > > The evaluation results are as follows:
> > >
> > > |            | CD   | F-Score | Lidar | Monocular Prior |
> > > |------------|------|---------|-------|-----------------|
> > > | StreetSurf | **0.52** | 56.70   | [x]   | [x]             |
> > > | Ours       | 0.55 | **61.54**   | [x]   |                 |
> > >
> > >
> > > We run StreetSurf with both monocular prior and lidar as input. While in our approach, we operate without relying on the monocular prior.
> > >
> > > We observe that our chamfer distance is higher than StreetSurf and F-Score is higher than StreetSurf. The reconstruction performance of observed geometry appears to be comparable.
> > >
> > > **Reply to Question 2:**
> > >
> > > We use SegFormer as the pretrained model for segmentation, and our results appear to be robust as a whole.
> > >
> > > Both our quantitative and qualitative results show that our method is stable with SegFormer.
> > >
> > > While it is inevitable that certain failure cases occur with small objects or object corners, these challenges are common across most segmentation methods. As segmentation techniques continue to evolve, our method is poised to benefit and improve alongside them.
> > >
> > > **Sum up:**
> > >
> > > We sincerely appreciate the valuable questions and constructive guidance from the reviewer to make this paper better. We will add discussions with figures about these two parts.
> > >
> > > We will notify the reviewer about the specific changes after our next paper revision submission.

---

> > > > ### Comment · Reviewer_SLKs · 2024-11-18
> > > > **Response to Rebuttal**
> > > >
> > > > Thanks for your reply! I keep my rating and raise my confidence.

---

> ### Author Response · Authors · 2024-11-19
>
> Dear Reviewer,
>
> We sincerely thank you once again for the valuable time and effort you have devoted to reviewing our work. Your insightful feedback has been instrumental in helping us improve the quality and clarity of our paper.
>
> In response to your suggestions, we have made the following revisions:
>
> 1. We have included both quantitative and qualitative experiments on the geometry of the "before unveiled" scene in Section C.1 of the supplementary PDF. We believe these additions will enrich the depth and comprehensiveness of the paper.
>
> 2. We have added a detailed discussion on semantic label supervision in Section D of the supplementary PDF, supported by the newly introduced Figure 13 for better illustration.
>
> Again, we deeply appreciate your constructive comments in order to make this paper better.
>
> **Update:**
>
> 3. We additionally extend the discussion about the static/dynamic decomposition in Section E of the supplementary PDF. (Corresponding to the Reviewer's feedback in Question 2 of the initial review)

---

### Official Review · Reviewer_orfH · 2024-11-03

**Soundness:** 3
**Presentation:** 2
**Contribution:** 3
**Rating:** 6
**Confidence:** 3

**Summary:**

The paper presents StreetUnveiler for reconstructing an empty street from in-car camera videos. It uses hard-label semantic 2DGS for representation, divides the scene into regions located by an alpha map, and introduces a time-reversal inpainting framework. Experiments show its effectiveness in reconstructing the empty street for autonomous driving applications.

**Strengths:**

- The use of 2D Gaussian combined with semantic information is reasonable. It allows for better identification and removal of objects by injecting semantic labels into 2D Gaussians.
- Experiments on street scene datasets demonstrate the effectiveness of the method.
- The proposed time-reversal framework inpaints frames in reverse order and uses later frames as references for earlier frames. This enhances the temporal consistency of the inpainting results, making the reconstructed street more realistic and consistent across different views.

**Weaknesses:**

- The 2DGS reconstruction and optimization processes might be computationally complex and costly, especially for large-scale street scenes, leading to long training and inference times. The author may consider other faster reconstruction baselines.
- The method has limited ability to handle dynamic objects. Although some approaches for simple dynamic cases are proposed, more improvements are needed for complex dynamic scenes.

**Questions:**

In fact, there are many methods that directly predict the masks of dynamic objects, and these are usually more reliable than the masks obtained by optical flow. Additionally, in many complex situations (such as vehicles driving sideways at an intersection), can the prior advantage of time-reversal inpainting still be maintained?

---

> ### Author Response · Authors · 2024-11-18
>
> We respectfully acknowledge the reviewer orfH’s valuable contribution to our work to improve this paper. We’d be delighted if your concerns could be solved by our reply.
>
> Reply to Strengths:
>
> We appreciate the reviewer’s recognition of semantic-aware design and the enhancement of temporal consistency brought by the time-reversal framework.
>
> Reply to Weaknesses & Questions:
>
> 1. “The 2DGS reconstruction and optimization processes might be computationally complex and costly, especially for large-scale street scenes, leading to long training and inference times. The author may consider other faster reconstruction baselines.”
>
> We appreciate the reviewer’s comment. We utilize 2DGS for its accurate geometry property, which is essential for the application of mesh extraction. Meanwhile, we provide a computational analysis in Section B.4 and Table.4 of our supplementary PDF, and it takes 2-3 hours for reconstruction, which may be not unacceptably slow. The technique in some recent work, like InstantSplat, has the potential to further accelerate the reconstruction process into 30 minutes. However, its acceleration requires the accurate initialization of the scene, which would not be so easy for large-scale street scenes.
>
> Technically, our manuscript code was conducted before [this 2DGS acceleration upgrade](https://github.com/hbb1/diff-surfel-rasterization/pull/7), which improves performance by 30–40% through optimized global memory access in the kernel function. Enhancements like this continuously improve the efficiency of the reconstruction stage in our pipeline.
>
> 2. “The method has limited ability to handle dynamic objects. Although some approaches for simple dynamic cases are proposed, more improvements are needed for complex dynamic scenes.”
>
> We appreciate the reviewer’s concerns. As the first to investigate 3D inpainting with single-pass data in long street scenes, rather than simpler small-scale scenes, we focus on static scenarios. While this is already a challenging problem that existing baseline methods cannot address effectively, we provide a “Discussion on Dynamic Object Removal” in Section D of our supplementary PDF. Some explicit dynamic modeling may be introduced in future work to achieve better results in more challenging dynamic cases. However, our current inpainting strategy remains effective for static objects, which are not addressed by dynamic modeling approaches.
>
> 3. “In fact, there are many methods that directly predict the masks of dynamic objects, and these are usually more reliable than the masks obtained by optical flow.”
>
> We appreciate the reviewer’s comment. We agree that dynamic object detection would be a powerful tool for handling dynamic object removal if the segmentation results are robust enough. And would potentially be an important component for future work about handling more challenging dynamic cases.
>
> 4. “Additionally, in many complex situations (such as vehicles driving sideways at an intersection), can the prior advantage of time-reversal inpainting still be maintained?”
>
> We appreciate the reviewer’s question. In the case of “Driving sideways at an intersection”, the relative positional relationship between the video-capturing vehicle and the driving sideway vehicle would be similar to the video-capturing vehicle crossing the static vehicle. This positional relationship would be similar to the case in our experiment. (Firstly come near to each other and then get far away from each other) However, the actual effect may be affected by the dynamic challenges mentioned above and in Section. D of our supplementary PDF.
>
> We thanks again for your valuable questions.

---

> ### Author Response · Authors · 2024-11-19
>
> Dear Reviewer,
>
> We want to kindly follow up regarding our earlier response to your review. We would like to know whether our reply adequately addressed your concerns.
>
> We have revised our paper to include the potential solutions for dynamic object removal through dynamic object detection, provided in Section E of our supplementary PDF. (Corresponding to the feedback from the Questions of the initial review)
>
> Thank you again for your time and valuable insights, which have greatly contributed to improving our work. We deeply appreciate your guidance.

---

### Official Review · Reviewer_roan · 2024-11-03

**Soundness:** 3
**Presentation:** 3
**Contribution:** 1
**Rating:** 3
**Confidence:** 4

**Summary:**

This work aims to remove static movable objects (e.g., parked vehicles) in street view data for 3D reconstruction (e.g., 3DGS). The authors propose two main improvement to achieve better inpainting results: better inpainting mask, better reference for image inpainting.

Experiments conducted on Waymo and PandaSet datasets demonstrate better performance compared to image/video inpainting methods and other 3D inpainting approaches.

**Strengths:**

1. Clear writing and presentation, with a well-designed supplementary website
2. Two novel and effective design elements:
   * An improved method for generating inpainting masks that incorporates observations from other frames
   * Time-reversal inpainting that utilizes future frames as references

**Weaknesses:**

1. The contribution seems somewhat limited. While the two major contributions are effective, they appear to be incremental rather than groundbreaking. The work could serve as a baseline, but not much more that.

2. The performance metrics are not entirely convincing. Table 1 doesn't demonstrate significant advantages over other baselines. The video in the supplementary materials appears noticeably inferior to other NVS methods on Waymo (e.g., EmerNeRF, StreetSurf, StreetGaussian).

3. The benchmark methodology raises some questions (please correct me if I'm mistaken, as I haven't encountered this benchmark setting before). What are the ground truth images used for LPIPS? If all rendered images lack vehicles while the ground truth images typically include them, is this an appropriate measurement for LPIPS/FID?

4. I have reservations about the task itself: Is it truly necessary to remove static components from the scene? Static elements (e.g., trees, traffic signs, rubbish bin) are integral parts of the environment. Even if removal is desired, wouldn't it be more efficient to accomplish this through multi-pass data collection?




## updated response to the authors after rebuttal
I want to clarify that my use of 'this issue will kill the paper' is a common academic shorthand expressing concern about a critical technical issue that needs to be addressed.  My colleagues and I often use this phrase in academic discussions to highlight fundamental issues that could seriously affect a paper's contribution. If you're unfamiliar with this expression and find it harsh, I sincerely apologize – that wasn't my intention.

**Questions:**

please address the concern in the weakness session

---

> ### Author Response · Authors · 2024-11-18
>
> We respectfully acknowledge the reviewer roan’s valuable comments on our work to improve this paper. We’d be delighted if your concerns could be solved by our reply.
>
> Reply to Strengths:
> We appreciate the reviewer’s recognition of our paper’s novelty and effectiveness in the methodology’s designs.
>
> Reply to Weaknesses & Questions:
>
> 1. “The contribution seems somewhat limited. While the two major contributions are effective, they appear to be incremental rather than groundbreaking. The work could serve as a baseline, but not much more that.”
>
> We appreciate the reviewer’s comment. To the best of our knowledge, we are the first work that extends the 3D inpainting task from simple scenes to large-scale street environments with one-pass data, successfully reconstructing an empty street, which provides temporal consistency videos, good novel-view synthesis results, and good geometry. Our approach also provides a verified solution to temporal consistency over a long sequence, which is relatively understudied, as highlighted by reviewer SpDA. The results of our method can be readily integrated into a simulator for autonomous driving or a game engine.
>
> 2. “The performance metrics are not entirely convincing. Table 1 doesn't demonstrate significant advantages over other baselines. The video in the supplementary materials appears noticeably inferior to other NVS methods on Waymo (e.g., EmerNeRF, StreetSurf, StreetGaussian).”
>
> We appreciate the reviewer’s comment. In Table 1, our FID is only lower than the baseline SDXL, yet SDXL doesn’t maintain temporal consistency between different video frames. Further, SDXL-based per image inpainting solution cannot achieve good geometry and does not support novel view synthesis.
>
> The task settings of the other NVS methods mentioned by reviewer(EmerNeRF, StreetSurf, StreetGaussian) are not the same as those in our task. Our paper focuses on a challenging unsupervised problem with one-pass data of a street scene, where **no ground truth data** exists for the "after unveiled" street. It’s not the same between “reconstructing the scene with ground truth data” and “reconstructing the background that’s completely blocked by static objects”. The works the reviewer mentioned here are mainly focusing on reconstructing the original fully-observed scene. Thus, these methods cannot be applied to our current settings. That's why we can not compare our method with them.
>
> 3. “The benchmark methodology raises some questions (please correct me if I'm mistaken, as I haven't encountered this benchmark setting before). What are the ground truth images used for LPIPS?”
>
> We appreciate the reviewer’s comment. Our task setting doesn’t have ground truth images, and that’s why this task is very challenging.
>
> Following well-established baselines(SpIN-NeRF, and [1,2,3]), we evaluate with LPIPS and FID. Each frame of the output video is paired with the corresponding frame from the original training video to compute the LPIPS. We use the image collections of the output video and original training video to compute FID. LPIPS directly compares the visual similarity between two images, focusing on perceptual quality differences. FID measures the similarity between the distributions of generated images and real images.
>
> 4. “If all rendered images lack vehicles while the ground truth images typically include them, is this an appropriate measurement for LPIPS/FID?”
>
> Actually, there are also some other long-standing tasks without ground truth for evaluation, like image/video generation, and some representative works in this direction, including StyleGan and DDPM, also use FID for evaluation. Their experiment is also an appreciated measurement when the ground truth images are lacking.
>
> [1] Silvan Weder, Guillermo Garcia-Hernando, Aron Monszpart, Marc Pollefeys, Gabriel Brostow, Michael Firman, and Sara Vicente. Removing objects from neural radiance fields. In Proceedings of the IEEE/CVF Conference on Computer Vision and Pattern Recognition (CVPR), 2023.
>
> [2] Zhiheng Liu, Hao Ouyang, Qiuyu Wang, Ka Leong Cheng, Jie Xiao, Kai Zhu, Nan Xue, Yu Liu, Yujun Shen, and Yang Cao. Infusion: Inpainting 3d gaussians via learning depth completion from diffusion prior. arXiv preprint arXiv:2404.11613, 2024.
>
> [3]Chieh Hubert, Changil Kim, Jia-Bin Huang, Qinbo Li, Chih-Yao Ma, Johannes Kopf, Ming-Hsuan Yang, and Hung-Yu Tseng. Taming latent diffusion model for neural radiance field inpainting. In European Conference on Computer Vision (ECCV), 2024a.

---

> > ### Author Response · Authors · 2024-11-18
> >
> > 5. “Is it truly necessary to remove static components from the scene? Static elements (e.g., trees, traffic signs, rubbish bin) are integral parts of the environment.”
> >
> > Thanks for sharing your concern. For the static elements, the reviewer mentioned (trees, traffic signs, rubbish bin). Generally, they can be split into two kinds of static components. One is temporarily placed in the scene(like cars, pedestrian, and the rubbish bin you mentioned); they are not part of the environment. Another is part of the environment(like trees and traffic signs). We can identify the targeted temporary static objects with our proposed semantic-aware approach, which is why semantic-aware is necessary in our pipeline.
> >
> > 6. “Even if removal is desired, wouldn't it be more efficient to accomplish this through multi-pass data collection?”
> >
> > Reconstructing the empty street scene with the multi-pass/multi-traversal data mentioned by the reviewer roan is not the same task as our one-pass task setting. Multi-pass data settings have its own challenges that are different from ours, like “diverse and complicated illumination over day and night”, “efficient training over dozens of multi-pass data”. We respect these methodological differences and distinct focus areas over different task settings.
> >
> > (Picking one single run in multi-pass data would be a one-pass case.)
> >
> > We thanks again for your valuable questions.

---

> > > ### Author Response · Authors · 2024-11-19
> > >
> > > Dear Reviewer,
> > >
> > > We want to kindly follow up regarding our earlier response to your review. We would like to know whether our reply adequately addressed your concerns.
> > >
> > > Thank you again for your time and valuable insights, which have greatly contributed to improving our work. We deeply appreciate your guidance.

---

### Official Review · Reviewer_SpDA · 2024-11-03

**Soundness:** 3
**Presentation:** 3
**Contribution:** 3
**Rating:** 8
**Confidence:** 4

**Summary:**

This paper introduces StreetUnveiler, a method to reconstruct empty street scenes from crowded video captured by moving vehicles. The key challenge is removing temporarily static objects (like parked cars and pedestrians) while maintaining temporal consistency over long trajectories. The main contributions are: 1) adapting 2DGS with hard-label semantic for scalable scene representation with explicit object removal, 2) using rendered alpha map to classify areas as fully observed, partially observed, or unobserved for creating inpainting masks, and 3) a time-reversal inpainting framework that maintains temporal consistency by using future frames to condition earlier frames. The paper shows better inpainting results compared to other off-the-shelf methods.

**Strengths:**

- I particularly like the idea of time-traversal inpainting. While single-image inpainting has been extensively explored, video-based inpainting (where temporal consistency is required) remains relatively understudied. Although the overall process isn't completely novel, I appreciate that the method is specifically tailored for driving videos with predominantly forward motion. Reference-based inpainting shows great promise for many applications and seems like a valuable direction to pursue.

- The paper is well-written, with clear motivation. The process, insights, and technical details are thorough and convincing. I especially appreciate the comprehensive related work section, which is very helpful for readers who don't necessarily work in inpainting or neural scene representation.

- The results are impressive, with both quantitative and qualitative improvements over baseline inpainting methods. The method appears to handle challenging cases well, particularly in maintaining consistency across long sequences.

**Weaknesses:**

- My only concern is that while the ultimate goal is temporally-consistent inpainting, the authors chose a more complex approach going through neural scene representation (2D/3DGS) and using rendered alpha masks to identify different region types (observable/unobservable) for generating inpainting masks. While this is interesting in principle, I wonder if a simpler pipeline could achieve similar goals - for example, establishing object correspondences across the video frames and then applying LeftRefill. I also note that LeftRefill is missing from the quantitative comparisons.

**Questions:**

I think the paper is well-written with a good application as it proposed an inpainting method that seems to work well to maintain temporal consistency.

---

> ### Author Response · Authors · 2024-11-18
>
> We respectfully acknowledge the reviewer SpDA’s valuable contribution to our work to improve this paper. We’d be delighted if your concerns could be solved by our reply.
>
> Reply to Strengths:
>
> We appreciate the reviewer’s recognition of our paper’s methodology tailored for driving videos and our results.
>
> Reply to Weaknesses & Questions:
> 1. “I wonder if a simpler pipeline could achieve similar goals - for example, establishing object correspondences across the video frames and then applying LeftRefill”
>
> We appreciate the reviewer’s question, while the 3D representation obtained by our method has some more applications. For example, an empty street’s geometry can be extracted by our final 3D representation, as placed as an example in Figure 16 and Figure 17 of our supplementary PDF, and such an empty scene can be integrated into a simulator for autonomous driving or a game engine. Further, Novel view synthesis is also a function that a simpler 2D video pipeline won’t achieve.
>
> 2. “I also note that LeftRefill is missing from the quantitative comparisons.”
>
> We thank the reviewer for pointing this out. We additionally conduct a quantitative comparison of LeftRefill. Since LeftRefill requires an image as a reference, LeftRefill can’t be naturally run as unconditional inpainting methods like LAMA and SDXL in Table 1.
>
> We adapt LeftRefill with the 10th future frame as a condition and use the mask obtained after our reconstruction stage.
>
> |     | Waymo |     | Pandaset |     |
> | --- | --- | --- | --- | --- |
> |     | LPIPS ↓ | FID ↓ | LPIPS ↓ | FID ↓ |
> | LeftRefill | 0.227 | 135.421 | 0.288 | 168.112 |
> | **Ours** | **0.216** | **127.581** | **0.261** | **155.527** |
>
> We observe that the evaluated LPIPS and FID of LeftRefill are all lower than ours on both Waymo and Pandaset.
>
> This naive reverse inpainting with LeftRefill will cause the red mask region in Figure 4 of our paper to be inpainted with the future frame as a reference, but some regions in the red mask are not visible in the future frame. This will lead to the wrong content generated by LeftRefill. We will revise this paper with an additional discussion about this with both qualitative and quantitative comparisons, which we believe will further enhance this paper.
>
> We thanks again for your valuable questions.

---

> > ### Author Response · Authors · 2024-11-19
> >
> > Dear Reviewer,
> >
> > We sincerely appreciate the time and effort you have dedicated to reviewing our paper. Your valuable feedback has significantly contributed to improving its quality.
> >
> > In response to our last comment, we have made revisions and included a new section, Section B.3, in the supplementary PDF. This section provides a detailed comparison with LeftReill, featuring both qualitative and quantitative analyses.
> >
> > We are grateful for your constructive insights and hope these additions address your concerns effectively.
> >
> > Again, we express our deep respect and gratitude for your valuable feedback to make this paper better.

---

> > ### Comment · Reviewer_SpDA · 2024-11-22
> > **Response to Rebuttal**
> >
> > Thank you for your response! Although the LPIPS improvement is much smaller than I expected based on the qualitative results shown in the paper (e.g., I don't think there's much difference between LPIPS of 0.227 and 0.216 -- maybe it's a weakness of the metrics itself), I think the paper offers a positive contribution to the community. I will keep my rating.

---

### Comment · Reviewer_roan · 2024-11-25
**Discussion on the contribution**

Dear fellow reviewers and authors,

I would like to start a public discussion to express some concerns about this paper:

1, Regarding Contribution: The key idea of building 3D scene representations to render unoccluded views by leveraging multi-view observations has been well explored in prior work. Dozens of methods have demonstrated similar capabilities in scene decomposition and object removal, e.g.:

1. https://light.princeton.edu/publication/neural-scene-graphs/
2. https://waabi.ai/unisim/
3. https://research.zenseact.com/publications/neurad/
4. https://emernerf.github.io/
5. https://zju3dv.github.io/street_gaussians/

The main extension in this paper appears to be the addition of an inpainting network for fully occluded regions that aren't visible in any source views. While this addresses a practical issue, the technical contribution seems incremental, and still have obvious artifacts on large invisible regions.

2, Regarding quality: for visible regions: The re-rendered results show noticeably lower quality compared to optimization-based approaches, exhibiting more blur than the methods mentioned above (e.g streetgaussian).
For occluded regions: There are clear artifacts in the inpainting results, For example, in scene2/data_vs_ours.mp4 at timestamp 00:02, where the removal of the SUV on the left leaves visible distortions. Similar issues are present throughout other example videos.

While I acknowledge that the approach shows effectiveness in achieving its stated goals, the combination of incremental technical novelty and quality limitations raises questions about the significance of the contribution.

I welcome thoughts from other reviewers and the authors on these points.

---

> ### Author Response · Authors · 2024-11-26
>
> We thank the reviewer roan for sharing the concerns.
>
> We hope our reply can address your questions:
>
> 1. Regarding Contribution: The key idea of building 3D scene representations to render unoccluded views by leveraging multi-view observations has been well explored in prior work. Dozens of methods have demonstrated similar capabilities in scene decomposition and object removal.
>
> We appreciate the reviewer for sharing the concern. First of all, we would like to point out that the reviewer misunderstood our work.  The goals of your mentioned paper and that of our papers are also different. We elaborate on each work about its setting you mentioned here:
>
> (1) “Neural Scene Graphs for Dynamic Scenes”:  This paper decomposes dynamic scenes with neural rendering. Such work cannot remove the static contents (like static cars in the scene)
>
> (2) “UniSim: A Neural Closed-Loop Sensor Simulator”: This paper composites the static background and dynamic actors in the scene to simulate data from new viewpoints. It is worth noting that the non-essential static elements, like stopping vehicles, are also part of the static background in their setting. This paper cannot remove these non-essential static elements.
>
> (3) “NeuRAD: Neural Rendering for Autonomous Driving”: This paper aims to design a robust novel view synthesis method tailored to dynamic AD data. It cannot recover a street without the stopping cars/pedestrians if stopping cars/pedestrians exist in the scene.
>
> (4) “EmerNeRF: Emergent Spatial-Temporal Scene Decomposition via Self-Supervision”: This paper decomposes dynamic scenes into static and dynamic components. (Note: The non-essential elements we wish to remove are not disentangled from its static components.) It still cannot remove those non-essential static elements.
>
> (5) “Street Gaussians: Modeling Dynamic Urban Scenes with Gaussian Splatting”: This paper models the dynamic urban streets for autonomous driving scenes. It is worth noting that the non-essential elements we wish to remove are not disentangled from their static components in this paper. Consequently, the stopping cars/pedestrians cannot be removed in this paper.
>
> In summary, the goals of the papers you mentioned are totally different from ours. The works mentioned above mainly focus on how to decompose the dynamic and static scenes, while we aim to remove non-essential static elements from the scene with single-pass data.
>
> We would like to restate that, to the best of our knowledge, we are **the first to scale up the static object removal tasks to the long sequences data**.
>
> 2. “Regarding quality: for visible regions: The re-rendered results show noticeably lower quality compared to optimization-based approaches, exhibiting more blur than the methods mentioned above (e.g streetgaussian). For occluded regions: There are clear artifacts in the inpainting results, For example, in scene2/data_vs_ours.mp4 at timestamp 00:02, where the removal of the SUV on the left leaves visible distortions. Similar issues are present throughout other example videos.”
>
> Thank you for sharing your concern. We have elaborated on the difference in terms of the problem settings with methods like StreetGaussian.
>
> The task setting of optimization-based NVS methods (e.g streetgaussian) focuses on modeling both static and dynamic elements of the scene **with ground-truth data**. While our method aims to recover the completely occluded regions(like the ground under the vehicle), **it lacks the corresponding ground-truth data**. **The task settings are different for our method and the other methods you mentioned**. Thus, **it’s unfair and improper to compare methods in two different settings**. Thus, it is improper to mention that “The re-rendered results show noticeably lower quality compared to optimization-based approaches, exhibiting more blur than the methods mentioned above (e.g streetgaussian)” in your comments. Further, our method is the first work in the single-pass-based object removal in long-trajectory videos, and there is still some room for our method to improve in the future.

---

> > ### Comment · Reviewer_roan · 2024-11-26
> >
> > Thanks the authors for the quick response, but this does not really address my concerns
> > 1.
> > > In summary, the goals of the papers you mentioned are totally different from ours. The works mentioned above mainly focus on how to decompose the dynamic and static scenes, while we aim to remove non-essential static elements from the scene with single-pass data.
> >
> > Those works can "Inpaint" dynamic objects, they just don't claim  this as a contribution  as it's a very straightforward application.
> > Furthurmore, BlockNeRF https://waymo.com/research/block-nerf/ can do static object removal a while ago without emphasizing it's an inpainting work.
> >
> > While I agree this work can furthur remove non-essential static elements in single-pass, but the artifacts is still quite obvious and improvement seems marginal.
> >
> >
> > 2.
> >
> > >  it lacks the corresponding ground-truth data
> >
> > The emphasis of groundtruth-free on this work doesn't make much sense to me, as this work still requires pretrained segmentation model. Similiarly, streetgaussian can utilize pretrained image3D detection model to get the dynamic boxes.   Moreover, methods like EmerNeRF doesn't require labels or supervised pretrained model
> >
> > 3.
> > >  "The re-rendered results show noticeably lower quality compared to optimization-based approaches"
> >
> > My primary concern is that in fully visible regions (where no inpainting is required), the rendering quality appears to be noticeably lower than optimization-based approaches like StreetGaussians. Could you elaborate on why this gap exists, given that these regions should be reconstructible from direct observations?
> >
> > 4.
> > Lasty, you miss the citation for this work: Inpaint3D: 3D Scene Content Generation using 2D Inpainting Diffusion
> >
> > I appreciate your effort in the work and rapid response, but I believe these points deserve further discussion to better establish the paper's positioning and contributions.

---

> > > ### Author Response · Authors · 2024-11-26
> > >
> > > Dear reviewer, here is our reply:
> > >
> > > > 1. “Those works can "Inpaint" dynamic objects, they just don't claim this as a contribution as it's a very straightforward application. Furthurmore, BlockNeRF https://waymo.com/research/block-nerf/ can do static object removal a while ago without emphasizing it's an inpainting work.”
> > >
> > > Thanks for your question. We want to point out that BlockNeRF is a **Multi-pass** setting and it collects data in San Francisco over 3 months and collects 2.8 million images, While our work focuses on a **single-pass** setting. Our paper doesn’t share the same problem setting with BlockNeRF, while single-pass data is easier to collect than the multi-pass setting in BlockNeRF.
> > >
> > > > 2. “The emphasis of groundtruth-free on this work doesn't make much sense to me, as this work still requires pretrained segmentation model. Similiarly, streetgaussian can utilize pretrained image3D detection model to get the dynamic boxes. Moreover, methods like EmerNeRF doesn't require labels or supervised pretrained model”
> > >
> > > Thanks for sharing your opinion. We would like to point out that StreetGaussian and EmerNeRF are not aiming at the 3D inpainting of the static non-essential elements. These works don’t share the same setting as our problem setting. **The groundtruth-free refers to that we don’t have the groundtruth for the empty street.** The pretrained models mentioned are either for semantic segmentation or object detection, but not for our task. The discussion of whether a pretrained model is used or not will not change the fact of the groundtruth-free setting of our problem.
> > >
> > > > 3. “My primary concern is that in fully visible regions (where no inpainting is required), the rendering quality appears to be noticeably lower than optimization-based approaches like StreetGaussians. Could you elaborate on why this gap exists, given that these regions should be reconstructible from direct observations?”
> > >
> > > Thank you for sharing your concern.
> > >
> > > First of all, the settings of the two works are totally different. StreetGaussian focuses on reconstructing the dynamic urban streets, while ours is recovering the fully invisible regions. The settings of these two works are orthogonal. From the engineering aspect, we definitely can borrow the dynamic modeling or other techniques from StreetGaussians to further improve the quality of NVS, but it is beyond the study scope of the paper. **We further restate that the goal of our work is removing the non-essential static objects.** Thus it is no need and unfair to compare with StreetGaussians.
> > >
> > >
> > > > 4. “Lasty, you miss the citation for this work: Inpaint3D: 3D Scene Content Generation using 2D Inpainting Diffusion”
> > >
> > > Thank you for pointing it out. We will revise our paper with this additional citation following your suggestions for a more comprehensive related work section.

---

> > > > ### Comment · Reviewer_roan · 2024-11-26
> > > >
> > > > Thank you for your responses. However, the concerns remain:
> > > >
> > > > - "BlockNeRF is a Multi-pass setting and it collects data in San Francisco over 3 months and collects 2.8 million images" --> The data you mentioned is a collection of 2 full districts, not just streets. WOD has millions of images collected over years. Multi-pass collection isn't something hard to do for self-driving scenarios.
> > > >
> > > > - "The groundtruth-free refers to that we don't have the groundtruth for the empty street" --> StreetGaussians doesn't have groundtruth for regions occluded by dynamic actors either.
> > > >
> > > >
> > > > - **Important** "We further restate that the goal of our work is removing the non-essential static objects." --> I understand that this work can remove static objects in a single pass , but the results show significant limitations:
> > > >     - Regions not requiring removal becomes blurrier;
> > > >     - Regions recoverable by StreetGaussians, your results are inferior;
> > > >     - Regions unique to your method's capabilities, the artifacts remain quite obvious.
> > > >
> > > >     I'm not an expert in this task and have never read such papers before, however based on my practical experience, combining StreetGaussians with a 2D inpainting network could likely achieve comparable or better results if tuned properly.
> > > >
> > > >     For these reasons, the contribution seem incremental rather than substantial.  **This is the key concern I would like to be addressed**
> > > > - Optional (Feel free to ignore this point): Has this work been previously submitted elsewhere? If so, what feedback did you receive and how was it addressed?

---

> > > > > ### Author Response · Authors · 2024-11-27
> > > > >
> > > > > Dear reviewer, here is our reply:
> > > > >
> > > > > 1. “ The data you mentioned is a collection of 2 full districts, not just streets. WOD has millions of images collected over years. Multi-pass collection isn't something hard to do for self-driving scenarios.”
> > > > >
> > > > > Thank you for your opinion. The discussion about whether multi-pass is easy or not is beyond the study scope of this paper, although it is common sense that collecting single-pass data is much easier and more economical. Please note that our paper focuses on the single-pass setting. We believe single-pass data would be a more general kind of data.
> > > > >
> > > > >
> > > > > 2. “StreetGaussians doesn't have groundtruth for regions occluded by dynamic actors either.”
> > > > >
> > > > > Thank you for your opinion. We would like to restate that the groundtruth for the regions corresponding to the static objects is unavailable, while the region occluded by dynamic factors in StreetGaussian can be seen in other frames.
> > > > >
> > > > > As a simple example, you can observe the static vehicles in “Decomposition results” from https://zju3dv.github.io/street_gaussians/ are not decomposed, while removing the static vehicles is our method’s focus.
> > > > >
> > > > > 3. “I understand that this work can remove static objects in a single pass , but the results show significant limitations”
> > > > >
> > > > > Thank you for your opinion. Again, we want to highlight that our core contribution is the time-reversal inpainting pipeline to maintain the consistency that is challenging in the long sequences.
> > > > >
> > > > > Our contribution is orthogonal to StreetGaussian. **Our contribution is not in conflict with StreetGaussian.** For better understanding, we exemplify that you mention “combining StreetGaussians with a 2D inpainting network”. The orthogonal combination would be “combining StreetGaussian with our time-reversal inpainting pipeline”. The naive “2D inpainting network” you mentioned can be replaced by our method for better consistency.
> > > > >
> > > > > 4. “Optional (Feel free to ignore this point): Has this work been previously submitted elsewhere? If so, what feedback did you receive and how was it addressed?”
> > > > >
> > > > > We would like to focus our discussion on the content of our paper. We kindly respect but refuse to answer this question.

---

> ### Comment · Reviewer_roan · 2024-11-27
>
> >  our core contribution is the time-reversal inpainting pipeline
>
> # **This claim could kill the paper**
>
> The regions that can *indeed* benefit from "time-reversal inpainting" are the regions that are visible in other frames, which can be inpainted in streetgaussian (and any other methods I mentioned)
>
> Streetgaussian, without any of these heuristics tricks, produce way better results in these regions.
>
> I would say the contribution lies in incorporating 2D inpainting techniques to handle non-essential object removal in 3D scene representations.  But both the methodology and results appear too preliminary.
>
> > The orthogonal combination would be “combining StreetGaussian with our time-reversal inpainting pipeline
>
> It is the authors' responsibility to implement solid baselines for comparison and demonstrate competitive results by building upon the best available methods.   While minor differences would be acceptable, the visual quality gap in the current results appears quite significant compared to the state-of-the-art.

---

> > ### Author Response · Authors · 2024-12-04
> >
> > We stand for respectful and friendly conversation. We’re only willing to reply to your questions and don’t want to get involved in any possible unfriendly conversations after we see the reviewer roan using **threatening and rude word like ‘kill’**.
> >
> > > 1. The regions that can indeed benefit from "time-reversal inpainting" are the regions that are visible in other frames, which can be inpainted in streetgaussian (and any other methods I mentioned)
> >
> > The argument “regions that are visible in other frames” is **a factual error** in the understanding of our work.
> >
> > As a very simple example, we aim to recover the regions under the stopping cars. This kind of occluded region **is absolutely invisible in all frames**. (Unless asking the owner of the stopping car to drive their car away before capturing the video.)
> >
> > > 2. “It is the authors' responsibility to implement solid baselines for comparison and demonstrate competitive results by building upon the best available methods.”
> >
> > The opinion that refers StreetGaussian as a baseline **is still a factual error** in the understanding of our work.
> >
> > StreetGaussian focuses on “dynamic-static decomposition”. Our setting focuses on recovering the region **absolutely invisible in all frames (the region under the stopping cars), which has no relationship with “dynamic-static decomposition”**.
> >
> > The areas that our paper wishes to recover under the parked cars are completely invisible, but not because of the impact of dynamic objects.
> >
> > We suggest the reviewer clearly understand StreetGaussian(dynamic-static decomposition) is not the same setting as ours(non-essential static objects removal), since we have explained it multiple times and provided tons of simple examples to clarify the difference.

---

### Meta-Review · Area_Chair_7pay · 2024-12-19

**Metareview:**

In this paper, the authors have proposed 3D StreetUnveiler to reconstruct an empty street from crowded observations based on semantic 2DGS. It first uses a rendered alpha map to locate unobserved regions, and creates a inpainting mask. Then, a time-reversal inpainting framework is proposed to enhance the temporal consistency. The paper is well-written and the contributions are recognized by most reviewers. There are some concerns raised by Reviewer roan which should be considered from my point of view. (1) A comprehensive comparison with StreetGaussian, e.g. the example in https://annoy449.github.io/output.mp4. (2) Some technical contributions of the paper are engineering and the novelty of the design should be highlighted. I agree that despite of these concerns, this work is still inspiring and will encourage more works in this topic. I would recommend a decision of accept as poster.

**Additional Comments On Reviewer Discussion:**

There were heated discussions during the rebuttal period. Initially reviewers raised concerns on the contributions, experiments, the insight of the design,  and the computational time of the proposed method. The authors have solved most of them, yet still have some issues to address as I mentioned in the metareview. I think this paper can be accepted though there is room for improvement.

---

### Decision · Program_Chairs · 2025-01-22

Accept (Poster)